# Greedy Bayesian Posterior Approximation with Deep Ensembles

**Aleksei Tiulpin**[*]                                                    *aleksei.tiulpin@oulu.fi*
*Research Unit of Medical Imaging, Physics and Technology*
*Faculty of Medicine,*
*University of Oulu, Finland*

**Matthew B. Blaschko**                              *matthew.blaschko@esat.kueleuven.be*
*Center for Processing Speech and Images*
*Department of Electrical Engineering*
*KU Leuven, Belgium*

**Reviewed on OpenReview:** *https://openreview.net/forum?id=P1DuPJzVTN*

## Abstract

Ensembles of independently trained neural networks are a state-of-the-art approach to estimate predictive uncertainty in Deep Learning, and can be interpreted as an approximation of the posterior distribution via a mixture of delta functions. The training of ensembles relies on non-convexity of the loss landscape and random initialization of their individual members, making the resulting posterior approximation uncontrolled. This paper proposes a novel and principled method to tackle this limitation, minimizing an $f$-divergence between the true posterior and a kernel density estimator (KDE) in a function space. We analyze this objective from a combinatorial point of view, and show that it is submodular with respect to mixture components for any $f$. Subsequently, we consider the problem of greedy ensemble construction. From the marginal gain on the negative $f$-divergence, which quantifies an improvement in posterior approximation yielded by adding a new component into the KDE, we derive a novel diversity term for ensemble methods. The performance of our approach is demonstrated on computer vision out-of-distribution detection benchmarks in a range of architectures trained on multiple datasets. The source code of our method is made publicly available at `https://github.com/Oulu-IMEDS/greedy_ensembles_training`.

## 1 Introduction

Estimation of predictive uncertainty is one of the most important challenges to solve in Deep Learning (DL). Applications in finance, medicine and self-driving cars are examples where reliable uncertainty estimation may help to avoid substantial financial losses, improve patient outcomes, or prevent fatal accidents (Gal, 2016). However, to date, despite rapid progress, there is a lack of principled methods that reliably estimate the predictive uncertainty of deep neural networks (DNNs).

Bayesian approaches to Machine Learning generally offer great benefits, which provide out-of-the-box features such as model selection (Immer et al., 2021), uncertainty quantification Wilson & Izmailov (2020), and incorporation of prior knowledge into the models (Fortuin, 2021). From a Bayesian standpoint, for a model $z$ trained on some data $\mathcal{D}$, there exist four main components: posterior $p(z \mid \mathcal{D})$, prior $p(z)$, likelihood $p(\mathcal{D} \mid z)$ and evidence $p(\mathcal{D})$. In this work we focus on the applications to uncertainty estimation, and thus are interested in a posterior distribution $p(z \mid \mathcal{D})$, assuming that it is multimodal (Wilson & Izmailov, 2020). This assumption is natural in the case of overparameterized models, such as DNNs.

---

[*]A part of this work was done at KU Leuven, and a part at Aalto University, Finland.

Numerous attempts have been made to develop Bayesian techniques for posterior approximation and uncertainty estimation in DL (Gal & Ghahramani, 2016; Lakshminarayanan et al., 2017; Ciosek et al., 2019; Maddox et al., 2019; Izmailov et al., 2020; Van Amersfoort et al., 2020; Wenzel et al., 2020b; He et al., 2020; Wilson & Izmailov, 2020). One of the most practical and empirically best-performing approaches is based on training a series of independent DNNs (Lakshminarayanan et al., 2017; Wilson & Izmailov, 2020; Ashukha et al., 2020; Lu et al., 2020; Wenzel et al., 2020b). The main method in this category, *Deep Ensembles* (DE) (Lakshminarayanan et al., 2017), is used as a reference approach in the context of this paper.

Recent studies, e.g. Wilson & Izmailov (2020) interpret ensembles as an approximation of predictive posterior. While this interpretation is correct from a Bayesian point of view, obtaining individual ensemble members via maximum a posteriori probability (MAP) estimation, as e.g. done in DE (Lakshminarayanan et al., 2017), may not lead to obtaining good coverage of the full support of the posterior distribution, and has arbitrarily bad approximation guarantees. For example, the resulting approximation can be poor in the case when the true posterior distribution is unimodal, skewed and long-tailed. In this work, we argue that enforcing coverage of posterior (i.e. ensemble diversity) is non-trivial, and needs a specialized principled approach. We highlight this graphically in Figure 1.

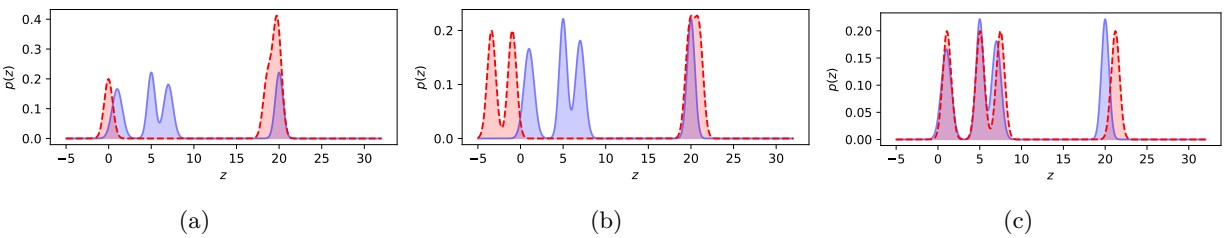

(a)  (b)  (c)

Figure 1: Illustration of how our method (c) approximates a 1D multimodal distribution $p(z)$ compared to a randomization-based mode picking (a), and naïve diversity training (b). Blue shows the true distribution and red – approximations. This figure can be reproduced using the provided source code.

Another important line of work in modern Bayesian DL (BDL) is a paradigm of performing Bayesian inference in the weight space. While distributions over weights induce distributions over functions (Wilson & Izmailov, 2020), it is rather unclear what the properties of such functions are, and whether Bayesian posteriors obtained in the weight space yield good quality approximations in the function space. For example, it is known that diverse weights do not necessary yield diverse functions, thus sampling from weight-based posteriors may yield poor quality uncertainty estimation, e.g. in detecting out-of-distribution (OOD) data (Garipov et al., 2018; Hafner et al., 2020).

Recent studies (Wang et al., 2019; D'Angelo & Fortuin, 2021) show the promise of particle optimization variational inference (POVI) done in the function space, however, the performance of those methods is not state-of-the-art due to the use of BNN weight priors. Specifically, in the BNN literature, enforcing prior only over weights and avoiding training techniques like batch-normalization Ioffe & Szegedy (2015) or mixup Zhang et al. (2018) is a de-facto standard, as these techniques have no Bayesian interpretation. In practice, a weight decay prior combined with batch normalization Ioffe & Szegedy (2015), data augmentation and dropout Gal & Ghahramani (2016) are often employed due to empirical improvement, and theoretical guarantees are traded for practical performance. Furthermore, particle-based function space VI requires training an ensemble of BNNs simultaneously, which is not only difficult to implement, but also requires extensive resources for parallelization.

**Summary of the contributions.** In this paper, we propose *a novel and principled* methodology for approximate function space posterior inference for DNNs. Contrary to the mainstream approach in BDL, which is based on defining a posterior distribution over the model parameters (Maddox et al., 2019), we take a functional view, which allows us to treat the problem of training ensembles as optimization over sets. Namely, selecting a set of functions to approximate the Bayesian posterior is fundamentally similar to problems such as facility location and set cover, which are frequently solved using submodular optimization (Krause et al.,

2008). Specifically, our contributions are:

1. We show that fitting a kernel density estimator to a distribution using an $f$-divergence is a cardinality-fixed non-monotone submodular maximization problem.

2. Inspired by the Random Greedy algorithm for submodular maximization (Buchbinder et al., 2014), we design a new method for the function space Bayesian posterior approximation via greedy training of ensembles with a justified coverage-promoting diversity term.

3. We demonstrate the effectiveness and competitiveness of our approach compared to DE in the OOD detection task on MNIST, CIFAR, and SVHN, benchmarks on different architectures and ensemble sizes. Furthermore, we show that our method can use state-of-the-art training techniques, compared to the existing Bayesian approaches, and yields state-of-the-art performance.

## 2 Preliminaries

### 2.1 Problem statement

Consider an ensemble to be parameterized by a set of functions $Z = \{z_m\}_{m=1}^{M} \subset V \subset \mathcal{F}$, where $V$ is a ground set, $\mathcal{F}$ is a class of continuous functions, and $z_m : \mathbb{R}^d \to \mathbb{R}^c$, with $d$ the dimensionality of the input data, and $c$ the dimensionality of the output. When training ensembles, we generally want to solve the following optimization problem:

$$\min_{Z \subset V, |Z|=M} \mathcal{R}(Z) - \Omega_{\lambda_M}(Z), \tag{1}$$

where $\mathcal{R}(Z) = \frac{1}{N} \sum_{i=1}^{N} \ell\left(\frac{1}{M} \sum_{m=1}^{M} z_m(x_i), y_i\right)$ is the empirical risk of the ensemble, $\ell : \mathcal{Y} \times \mathcal{Y} \to \mathbb{R}_+$ is a loss function, $\mathcal{D} = \{x_i, y_i\}_{i=1}^{N}$ is a training dataset of size $N$, and $\Omega_{\lambda_M}(Z)$ is some diversity-promoting term, with diversity regularization strength $\lambda_M$.

Empirical observations in the earlier works on ensembles (Lakshminarayanan et al., 2017; Wilson & Izmailov, 2020; Fort et al., 2019) have shown that one can simply ignore $\Omega_{\lambda_M}(Z)$ during optimization, and rely on non-convexity of the loss landscape, minimizing the risks of individual ensemble members. From a variational inference (VI) perspective (Zhang et al., 2019), this can be seen as a *mode-seeking* method, that is, the resulting posterior approximation method aims to put mass at the true posterior modes. Notably, it has been shown experimentally that every ensemble member may discover different modes of the posterior distribution in the function space $p(z|\mathcal{D})$ Fort et al. (2019).

Approximation of $p(z|\mathcal{D})$ is the ultimate aim of this paper, and we argue that randomization-based mode-seeking is insufficient to obtain a good quality approximation of $p(z|\mathcal{D})$, as this procedure does not maximize the coverage of the support of the posterior. In contrast, we aim to find an $\Omega_{\lambda_M}(Z)$ such that the posterior coverage is also maximized. Taking a VI perspective again (Zhang et al., 2019), $\Omega_{\lambda_M}(Z)$ needs to enforce that $\min_{Z \in V, |Z|=M} R(Z) - \Omega_{\lambda_M}(Z)$ has also *mean-seeking* behavior. That is, the approximation method should aim to cover as much of the true posterior support as possible while still discovering the high density modes.

### 2.2 A combinatorial view of ensemble construction

Having now defined the main criteria for (1), we highlight that the problem of constructing an ensemble can be seen from a combinatorial point of view. We therefore treat ensemble construction as subset selection from some ground set of functions, and introduce the main notions of submodular analysis, a powerful tool that enables the analysis of the optimization of set functions (Bach, 2013; Fujishige, 2005).

**Definition 1** (Submodularity). *A set function $g : 2^V \to \mathbb{R}$, for the power set of a base set $V$, is submodular if for all $A \subseteq B \subset V$ and $x \in V \setminus B$*

$$g(A \cup \{x\}) - g(A) \geq g(B \cup \{x\}) - g(B). \tag{2}$$

**Definition 2** (Supermodularity and modularity). *A set function is called supermodular if its negative is submodular, and modular if it is both submodular and supermodular.*

Consider now problem (1). Assuming that the loss function $\ell$ is convex, we can derive an upper-bound on the risk $\mathcal{R}(Z)$ using Jensen's inequality, and obtain a method, which generalizes DE

$$\min_{Z \subset V, |Z|=M} \frac{1}{M} \sum_{m=1}^{M} \mathcal{R}(z_m) - \Omega_{\lambda_M}(Z), \tag{3}$$

where $V$ is a ground set. In the context of neural networks, it is reasonable to consider the ground set $V$ containing all possible neural networks with a specific architecture and realizable in a computer.

If $M$ is fixed during optimization, $\frac{1}{M} \sum_{m=1}^{M} \mathcal{R}(z_m)$ contributes a positive modular term to the overall objective. Adding a positive modular function to any set function does not change its submodularity or supermodularity, thus we focus on $\Omega_{\lambda_M}(Z)$. A trivial approach would be to enforce pair-wise diversity by computing a norm of the pairwise differences between functions, i.e. setting $\Omega_{\lambda_M}(Z) = \lambda_M \sum_{i \neq j} \|z_i - z_j\|_*^2$. However, this is a cardinality-fixed submodular *minimization* problem. It is known that it is strongly NP-hard, i.e. there exists no general polynomial time approximation algorithm for it (Svitkina & Fleischer, 2011). The poor quality approximation of this approach is highlighted in Figure 1. We therefore conclude that the choice of $\Omega_{\lambda_M}(Z)$ has a direct impact on the approximabilty of the objective.

## 3 Submodular analysis of $f$-divergences

### 3.1 $f$-divergences are supermodular functions

**Main result.** We now consider the problem of approximating a Bayesian posterior via minimization of an $f$-divergence. Here, we specifically aim our optimization procedure to have both mode and mean-seeking behaviors, i.e. cover the posterior distribution as much as possible, ending up in its mode. We furthermore aim to obtain a polynomial time algorithm that yields a good quality approximation guarantee. In this paper, we leverage classic definitions of approximation algorithm and approximation guarantees.

**Definition 3** (Approximation algorithm and guarantees (Williamson & Shmoys, 2011)). *A $\gamma$-approximation algorithm for an optimization problem is a polynomial-time algorithm that for all instances of the problem produces a solution whose value is within a factor $\gamma$ of the optimal solution. $\gamma$ in this case is called approximation guarantee.*

Let us now formally introduce $f$-divergences.

**Definition 4** ($f$-divergence). *Let $f : \mathbb{R}^+ \to \mathbb{R}$ be a convex function such that $f(1) = 0$, $P_z$ and $Q_z$ be distributions on a measurable space $(\Omega, \mathcal{Z})$ admitting densities $p(z)$ and $q(z)$ with respect to a base measure $dz$. If $P_z$ is absolutely continuous with respect to $Q_z$, $f$-divergence between $P_z$ and $Q_z$ is defined as*

$$D_f(P_z || Q_z) = \int_{\mathcal{Z}} f\left(\frac{p(z)}{q(z)}\right) q(z) dz. \tag{4}$$

Consider some density $p(z)$ over continuous functions. We define $q_M(z) = \frac{1}{M} \sum_{m=1}^{M} K(z, z_m)$, where $K_m(z) := K(z, z_m)$ is a kernel centered at $z_m$ used in the density estimation of $p(z)$. In addition, we simplify our notation, always assuming the existence of a measurable space $(\Omega, \mathcal{Z})$ and the base measure $dz$.

**Theorem 1.** *Any $f$-divergence*

$$D_f(p || q_M) = \int f\left(\frac{p(z)}{\frac{1}{M} \sum_{j=1}^{M} K_j(z)}\right) \frac{1}{M} \sum_{m=1}^{M} K_m(z) dz \tag{5}$$

*between a distribution $p(z)$ and a normalized mixture of $M$ kernels with equal weights is supermoduular in a cardinality-fixed setting, assuming that $\max_{q_M} D_f(p(z) || q_M(z)) < \infty$.*

*Proof.* The proof is shown in Appendix A.1. □

For some fixed $p(z)$, minimization of (5) with respect to $q_M(z)$ is equivalent to a cardinality-constrained maximization of a non-monotone submodular function of $Z = \{z_1, \ldots, z_M\}$. Approximation guarantees for problems of this form are given for non-negative submodular functions (Buchbinder et al., 2014).

One can convert (5) to a non-negative function by defining:

$$F(Z) := -D_f(p||q_M) + C, \tag{6}$$

where $C = \max_{Z \subset V} D_f(p||q_M)$ is a pre-defined constant, which is important for understanding approximation guarantees, but does not need to be computed in practice. After the described transformation, which leads to (6), we obtain $F(Z)$, a *non-negative non-monotone* submodular function.

**Approximation guarantees for $f$-divergences.** In the context of submodular functions, there exists an inapproximability result, obtained in (Gharan & Vondrák, 2011), which states that no general polynomial time algorithm with guarantees better than 0.491 exists to solve a submodular maximization problem with constrained cardinality. We note that in practice, however, it might still be possible to obtain a better approximation factor than 0.491 for some specific types of divergences, as these guarantees are defined for *all instances* of the optimization problem (Definition 3).

Let us consider the approximation guarantees for (6), denoting by $q_M^*$ the optimal solution, and by $\hat{q}_M$ a solution found by some algorithm. For an approximation factor $\gamma$, we have

$$-D_f(p||\hat{q}_M) + C \geq \gamma(-D_f(p||q_M^*) + C). \tag{7}$$

Simple algebra shows that this implies

$$D_f(p||\hat{q}_M) \leq \gamma \min_{q_M \in A} D_f(p||q_M) + (1 - \gamma) \max_{q_M \in A} D_f(p||q_M), \tag{8}$$

where $A$ is a set of possible approximating mixture distributions of size $M$ constructed from the ground set. The derived result indicates that the upper bound on the approximate solution found by minimizing an $f$-divergence can be substantially dominated by $(1 - \gamma) \max_{q_M \in A} D_f(p||q_M)$ if the ground set $V$ is chosen poorly.

### 3.2 Greedy minimization of $f$-divergences

**Random greedy algorithm.** Although submodular optimization has natural parallel extensions and associated approximation guarantees, due to the simplicity of presentation, we focus in this paper on forward greedy selection and use Algorithm 1 for optimizing submodular functions. This algorithm has approximation guarantee of $\approx 1/e$ in general (Buchbinder et al., 2014). The only required step for this greedy algorithm is a computation of the marginal gain on the objective function $F(Z)$, i.e. $\Delta(z_k|Z) = F(Z \cup \{z_k\}) - F(Z)$.

---

**Algorithm 1** Random Greedy algorithm

---

1: **Input:** $V$ – Ground set
2: **Input:** $F$ – Arbitrary submodular function
3: **Input:** $M$ – Cardinality of the solution
4: $Z \leftarrow \emptyset$
5: **for** $m = 1$ **to** $M$ **do**
6:     $R \leftarrow \arg\max_{T \subset V \setminus Z : |T| = M} \sum_{z' \in T} \Delta(z'|Z)$
7:     $u_i \leftarrow \text{Uniform}(R)$
8:     $Z \leftarrow Z \cup \{u_i\}$
9: **end for**
10: **return** $Z$

---

**Marginal gain.** At each step of a greedy algorithm, a marginal gain $\Delta(z_k|Z) = F(Z \cup \{z_k\}) - F(Z)$ of adding a new element $z_k$ to an existing mixture $\sum_{j=1}^{k-1} K_j(z)$ is maximized. For $f$-divergences, we thus formulate the following proposition:

**Proposition 1.** *Consider $C = \max D_f(p||q_M)$, where $D_f(p||q_M)$ is an arbitrary $f$-divergence between some distribution $p(z)$ and a mixture of kernels $q_M(z) = \frac{1}{M} \sum_{j=1}^{M} K_j(z)$, and $D_f(p||q_M) < \infty$. Then, maximization of the marginal gain for the set function*

$$F(Z) = -\int f\left(\frac{p(z)}{\frac{1}{M}\sum_{j=1}^{M} K_j(z)}\right) \frac{1}{M} \sum_{m=1}^{M} K_m(z) dz + C, \tag{9}$$

*at step $k$ of a greedy algorithm corresponds to*

$$\arg\max_{z_k} \Delta(z_k|Z) = \arg\min_{z_k} \mathbb{E}_{z \sim K_k(z)} f\left(\frac{p(z)}{\frac{1}{M}\sum_{j=1}^{k} K_j(z)}\right). \tag{10}$$

*Proof.* The proof is shown in Appendix A.2. □

**The case of reverse KL divergence.** Having mean-seeking behavior is useful to fit the kernel density estimator using *forward* divergences. However, if one wants to optimize marginal gains, they need to have a mode-seeking behavior, which is achieved via optimizing *reverse* divergences (Zhang et al., 2019).

If we consider the generator for the reverse KL-divergence, $f(x) = -\log x$, the minimization becomes

$$\min_{z_k} \mathbb{E}_{z \sim K_k(z)} - \log p(z) + \log\left(\frac{1}{M} \sum_{j=1}^{k} K_j(z)\right). \tag{11}$$

We note that it is costly or even intractable to compute the expectation in (11), and we thus resort to the mean-field assumption. Specifically,

$$p(z) = p(z_1, \ldots, z_M) = \prod_{m=1}^{M} p(z_m), \tag{12}$$

where $z_m$ are variables that partition $p(z)$. In the case of arbitrary multimodal distributions with finite number of modes, such an assumption is well-justified, and also leads to a computationally tractable optimization procedure. Having this assumption in mind, we obtain the following point estimate of (11):

$$\min_{z_k} - \log p(z_k) + \log\left(\frac{1}{M} \sum_{j=1}^{k-1} K_j(z_k)\right). \tag{13}$$

## 4 Greedy approximation of Bayesian posterior for parametric functions

### 4.1 Objective function

**Derivation of the negative marginal gain** In this section, we consider parametric continuous functions $z_\theta : \mathbb{R}^d \to \mathbb{R}^c$, where $d$ – dimension of the input space, $c$ – dimension of the output space, and $\theta$ denotes the parameters determining the function. While our derivations in this section hold for any measurable function that satisfies this definition, we assume a DNN to be our model of choice. For DNNs it is natural to consider a ground set $V$ to be a set of all neural networks with a fixed architecture and a random initialization scheme realizable on a computer. We note that this is a very large, but finite set when considering fixed precision weights.

We now use Bayes' theorem and a mean-field approximation, similarly to (13). This allows us to express the multimodal posterior over a neural network by a product of posteriors of individual ensemble members. Specifically, the factorized posterior is expressed as

$$p(z_\theta|\mathcal{D}) \propto p(z_{\theta_1}, \ldots, z_{\theta_M}|\mathcal{D}) \propto \prod_{m=1}^{M} p(\mathcal{D}|z_{\theta_m})p(z_{\theta_m}), \tag{14}$$

where $\theta_m$ are parameters, $p(\mathcal{D}|z_\theta)$ the likelihood and $p(z_\theta)$ the prior; $p(\mathcal{D}|z_\theta) \propto \prod_{i=1}^{n} \exp(-\ell(z_\theta(x_i), y_i))$, and $p(z_\theta) \propto \exp(-\lambda\|\theta\|_2^2)$.

To leverage our earlier defined submodular maximization machinery, and optimize negative marginal gains derived in (13) for approximating (14), we define the kernel density components via generalized exponential kernels $K_j(z_\theta) \propto \exp(-\lambda_M d(z_\theta, z_{\theta_j})^2)$, where $\lambda_M$ is proportional to the kernel width and $d(\cdot, \cdot)$ is a distance measure between functions. Substituting the defined kernels and the factorized posterior defined in (14) into (13), we obtain the following objective to minimize at the $k^{th}$ greedy step of Algorithm 1:

$$J(\theta_k) = \underbrace{\mathbb{E}_{(x,y)\sim p(x,y)}\ell(z_{\theta_k}(x), y) + \lambda\|\theta_k\|_2^2}_{\text{Negative marginal gain on } \mathcal{R}(Z)} + \underbrace{\log \sum_{j=1}^{k-1} \exp\left(-\frac{\lambda_M}{M}d(z_{\theta_k}, z_{\theta_j})^2\right)}_{\text{Negative marginal gain on } \Omega_{\lambda_M}(Z)}, \tag{15}$$

which is similar to the negative marginal gain on our originally defined high-level ensemble training objective (1), except that the diversity term $\Omega$ has a different form based on our submodular $f$-divergence optimization.

**Sampling-based approximation of the diversity term.** When minimizing (15), one needs to be able to compute the diversity term $\Omega = \log \sum_{j=1}^{k-1} \exp(-\lambda_M d(z_\theta, z_{\theta_j})^2)$, which is derived from a kernel density estimator we aim to fit to the true posterior. We note that this needs to be done *in the function space*, which makes this computation non-trivial.

We earlier defined $K_j(z)$ to be an individual kernel in a mixture $\frac{1}{M}\sum_{j=1}^{M} K_j(z)$. In order to be able to use the $f$-divergence, the individual components $K_j(z)$ must be density functions centered at $z_j$, which implies the need of a notion of similarity or the existence of a function norm, which are known to be NP-hard to compute for neural networks with depth greater than 3 (Rannen-Triki et al., 2019). We note that this intrinsic hardness result applies to *all* methods that define a meaningful posterior distribution in function space through a kernel density estimator. We thus use here a method from (Rannen-Triki et al., 2019) to approximate $\|z\|_2^2$, via i.i.d. samples $x_i \sim P^*$, where $P^*$ is a weighting distribution, which is required to ensure that Monte Carlo integration yields a reasonable approximation. This leads to the following sampling-based approximation of the diversity term:

$$\log \sum_{j=1}^{k-1} \exp\left(-\frac{\lambda_M}{M}\mathbb{E}_{x\sim p^*(x)}\|z_{\theta_k}(x) - z_{\theta_j}(x)\|_2^2\right). \tag{16}$$

## 4.2 Practical implementation

**Computing the diversity term** To this point, we defined all the main components of our method, except the weighting distribution in (16). The desirable behavior, which we expect an ensemble to exhibit, is that it must be uncertain on the OOD data and certain in the regions where the training data are available. This implies that $p^*(x)$ *must include* OOD samples. One can use OOD data in training explicitly, however, in our work, resort to a setting when OOD data are unknown. Specifically, we use a simple heuristic, which fits a Gaussian to every data dimension with the variance $\times 5$ larger than the variance of the data. We specify further details about this in Appendix B.1.

**The resulting algorithm** The resulting, computationally tractable optimization algorithm for ensembles, which minimizes negative marginal gains (10), is shown in Algorithm 2. For simplicity, we omit the snapshot selection step, i.e. early stopping.

We note that in Algorithm 1 line 7 we require a uniform random sampling over the top $M$ functions. In practice on a computer, the set of functions are parameterized by fixed precision floating point numbers, and that there are a large number of functions differing by a single lowest significant bit of one network weight, effectively achieving the same value of the negative marginal gain up to measurement error (recall the NP-hardness of the diversity term and its approximation by Monte Carlo integration). Therefore, the randomization introduced by Algorithm 1 line 7 is lower than the combination of optimization error and approximation error from Monte Carlo integration. There is therefore no gain from performing multiple optimizations and we replace the uniform random selection step with a single stochastic optimization:

At each $k^{th}$ greedy step, we change the seed (`set_seed`$(\cdot)$ in Algorithm 2) of the random number generator before initializing the model and optimize a neural network using stochastic gradient descent from random initialization. The final computational performance improvement can be obtained by storing the evaluations $z_j(x_i) \ \forall j = 1, \ldots, k-1$ in memory before executing each $k^{th}$ step.

We report here also one practical trick, which we found important during training. Specifically, freezing the batch normalization layers (Ioffe & Szegedy, 2015) before computing the diversity term turned out to help the convergence substantially.

---

**Algorithm 2** $\mathcal{O}(k)$ Random Greedy-based algorithm for training ensembles of neural networks. `set_seed`$(\cdot)$ sets the random seed. The $\arg\min_\theta$ on line 10 is achieved by stochastic gradient descent with random weight initialization.

---

1: **Input:** $V$ – Set of all neural networks specified by some architecture
2: **Input:** $M$ – Cardinality of the solution
3: **Input:** $p(x, y)$ – Training data / empirical distribution
4: **Input:** $p^*(x)$ – Weighting distribution for the diversity term
5: **Input:** $s$ – Initial random seed
6: $\theta_1 \leftarrow \arg\min_{\theta_m} \mathbb{E}_{(x,y) \sim p(x,y)} \ell(z_{\theta_1}(x), y) + \lambda\|\theta_1\|_2^2$
7: $Z \leftarrow \{z_{\theta_1}\}$
8: **for** $m = 2$ **to** $M$ **do**
9: $\quad$ `set_seed`$(s + m - 1)$;
10: $\quad \theta_m \leftarrow \arg\min_\theta \mathbb{E}_{(x,y)} \ell(z_\theta(x), y) + \lambda\|\theta\|_2^2 + \log \sum_{j=1}^{m-1} \exp\left(-\frac{\lambda_M}{M} \mathbb{E}_{x^* \sim p^*(x)} \|z_\theta(x^*) - z_{\theta_j}(x^*)\|_2^2\right)$
11: $\quad Z \leftarrow Z \cup \{z_{\theta_m}\}$;
12: **end for**
13: **return** $Z$

---

## 5  Related work

**Randomization-based ensembles.** Generally, ensembles have been studied in Machine Learning over several decades for different classes of models (Hansen & Salamon, 1990; Breiman, 1996; Freund & Schapire, 1997; Lakshminarayanan et al., 2017). One can distinguish several main methods for diverse ensemble construction (Pearce et al., 2020): randomization of training initialization and hyperparameters (Hansen & Salamon, 1990; Wenzel et al., 2020b; Wilson & Izmailov, 2020; Zaidi et al., 2021), bagging (Breiman, 1996; 2001), boosting (Freund & Schapire, 1997), and explicit diversity training (Kuncheva & Whitaker, 2003; Ross et al., 2020; Yang et al., 2020; Brown et al., 2005; Kariyappa & Qureshi, 2019; Sinha et al., 2021; Melville & Mooney, 2005). Randomization-based ensemble construction has shown good results in in-domain uncertainty (Ashukha et al., 2020) estimation, but also in the detection of OOD data (Lakshminarayanan et al., 2017).

**Submodular ensemble pruning and greedy ensemble construction** Submodularity in ensemble learning has previously been discussed in the context of ensemble pruning (Sha et al., 2014). The goal of ensemble pruning is to trim a large ensemble of models so that the accuracy of the ensemble remains the same. We note that recently, inspired by the submodular pruning approach, a greedy algorithm has been applied to randomization based ensembles (Wenzel et al., 2020b; Zaidi et al., 2021). However, we also note that neither of these works approached the problem of ensemble construction from a Bayesian posterior approximation

point of view. However, they provide extensive experimental evidence on the plausibility of greedy approach. Our work sheds light on *why* in Wenzel et al. (2020b); Zaidi et al. (2021) ensembles constructed greedily worked well in OOD detection tasks.

**Diversity-promoting regularization for ensemble training.**   The ensemble literature contains a line of work focusing on explicitly promoting regularization in ensembles (Kuncheva & Whitaker, 2003; Ross et al., 2020; Yang et al., 2020; Brown et al., 2005; Kariyappa & Qureshi, 2019; Sinha et al., 2021; Melville & Mooney, 2005; Havasi et al., 2021). In terms of promoting diversity outside training data, the closest work to ours is (Ross et al., 2020; Rame & Cord, 2021), and in terms of the form of diversity regularization it is (Kariyappa & Qureshi, 2019). However, none of these works takes the perspective of approximating the Bayesian posterior. Another limitation of most of these approaches is that they use either out-of-distribution data in training, adversarial examples, or expensive generative models, thus making those methods difficult to scale to large datasets.

**POVI**   A problem of learning diverse ensembles can be seen from a POVI perspective: in particular, Stein Variational Gradient Descent (SVGD) (Wang & Liu, 2019). SVGD aims to learn a diverse set of functions $Z$, approximating arbitrary distributions via a set of particles using a reverse KL divergence. This approach is similar to ours, however, we tackle the problem of optimizing a *general $f$-divergence* in the function space. Furthermore, to our knowledge, the present work is the first that takes a submodular minimization perspective in BDL.

Another line of work in the POVI family also considers greedy approximations (Futami et al., 2019; Jerfel et al., 2021). In both of these works, the authors perform particle based inference by also optimizing the kernel widths of the KDE, which is different to our method. Furthermore, these works do not tell what the diversity term for ensemble training should look like, and how to compute it in practice.

**Function space POVI**   Conventionally, Bayesian inference in DL is thought of in the weight space (MacKay, 1992; Blundell et al., 2015; Gal & Ghahramani, 2016; Izmailov et al., 2020; Wilson & Izmailov, 2020; Pearce et al., 2020; Wenzel et al., 2020a). However, recent studies point out that despite the fact that simple priors over weights may imply complex posteriors in the function space, the connection between two of these is difficult to establish (Sun et al., 2019; Hafner et al., 2020).

Recent papers on function-space POVI (Wang et al., 2019; D'Angelo & Fortuin, 2021) point out that one can do Bayesian inference in the function space by optimizing an objective function with a repulsive term. Notably, they draw connection to the reverse KL-divergence minimization, and we thus consider our method to be closely connected to function space POVI.

## 6  Experiments

### 6.1  Setup

**Datasets and models.**   We ran our main experiments on CIFAR10, CIFAR100 (Krizhevsky, 2009) and SVHN (Netzer et al., 2011) in-distribution datasets. Our OOD detection benchmark included CIFAR10, CIFAR100, DTD (Cimpoi et al., 2014), SVHN (Netzer et al., 2011), LSUN (Yu et al., 2015), TinyImageNet (Le & Yang, 2015), Places 365 (Zhou et al., 2017), Bernoulli noise images, Gaussian noise, random blobs image, and uniform noise images. The composition of the benchmark was inspired by the work of Hendrycks et al. (2019). We excluded the in-distribution datasets for each of the settings, resulting in a total of 10 OOD datasets for each in-distribution dataset. The full description of the benchmark is shown in Appendix B.2.

The experiments were conducted using ResNet164 (pre-activated version; denoted as PreResNet164) (He et al., 2016), VGG16 (with batch normalization (Ioffe & Szegedy, 2015); denoted as VGG16BN) (Simonyan & Zisserman, 2015), and WideResNet28x10 (Zagoruyko & Komodakis, 2016). All our models in the ensembles were trained for 100 epochs using PyTorch (Paszke et al., 2019), each ensemble on a single NVIDIA V100 GPU. In the case of CIFAR and SVHN experiments, we trained ensembles of size $M = 11$, and report the results across 5 different random seeds. For the CIFAR experiments, we selected $\lambda_M \in$

$\{0.001, 0.005, 0.01, 0.05, 0.1, 0.5, 1, 3, 5, 7, 10\}$. In addition to the CIFAR and SVHN experiments, we used MNIST (LeCun et al., 1998) with ResNet8. The details of those experiments are shown in Appendix B.2.

**Model selection and metrics.**    We used mutual information (MI) between the distribution of the predicted label $\hat{y}$ for the point $\hat{\mathbf{x}}$ and the posterior distribution over functions $p(f|\mathcal{D})$, to evaluate the *epistemic uncertainty* (Malinin & Gales, 2018; Depeweg et al., 2018) and reported the area under the ROC curve (AUC) and area under the precision-recall (PR) curve, i.e. average precision (AP) to quantify the OOD detection performance. Furthermore, we computed the false positive rate at 95% true positive rate (FPR95). Details on the computation of epistemic uncertainty can be found in Appendix B.3.

## 6.2   Results

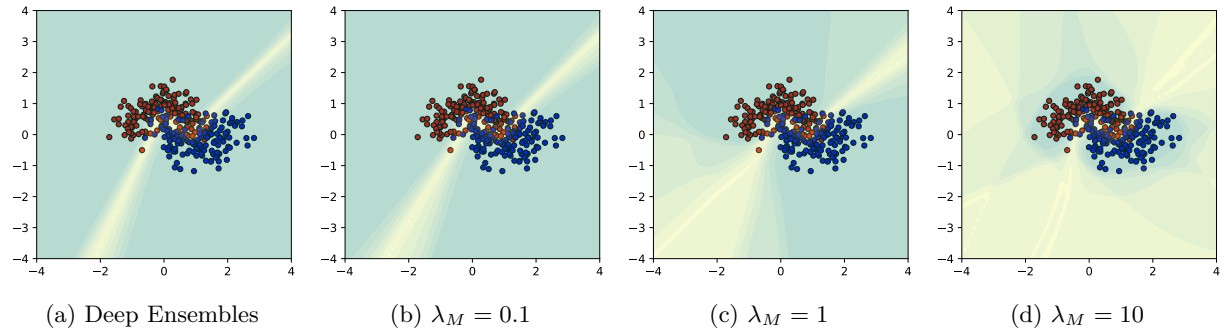

(a) Deep Ensembles          (b) $\lambda_M = 0.1$          (c) $\lambda_M = 1$          (d) $\lambda_M = 10$

Figure 2: Uncertainty of an ensemble of two layer neural networks on a two moons dataset (size $M = 11$). Compared to DE, which is uncertain only close to the decision boundary, our method yields the desired behavior – the further we move from training data, the higher uncertainty is. Such behavior is controlled by the diversity regularization coefficient $\lambda_M$.

**Illustrative examples.**    Figure 2 illustrates how our method performs on the two moons dataset. Here, we used a two-layer fully-connected network with ReLU activations (Fukushima, 1988). Having a high $\lambda_M$ is important to obtain good uncertainty estimation. As expected, compared to our method, the DE method does not explicitly maximize the coverage of the posterior, and thus fails to be uncertain outside the training data.

**Out-of-distribution detection.**    We present aggregated results for all the models and in-distribution datasets in Table 1. It is clear that on average (across OOD datasets), our method is substantially better than DE. This holds for all the architectures and in-distribution datasets. We show the expanded version of all the OOD detection results in Appendix C.3. An example of these results is shown in Figure 3 for all the models trained on CIFAR 100. Here, one can see that our method is at least similar to DE, and substantially better overall in terms of AUC and AP metrics. Finally, some examples of OOD detection by both DE and our method are shown in Figure 4. Here, we computed optimal thresholds for each of the methods by optimizing the trade-off between the true positive and true negative rates.

## 7   Discussion

In this paper, we have introduced a novel paradigm for Bayesian posterior approximation in Deep Learning using greedy ensemble construction via submodular optimization. We have proven a new general theoretical result, which shows that minimization of an $f$-divergence between some distribution and a kernel density estimator has approximation guarantees, and can be done greedily. We then derived a novel coverage promoting diversity term for ensemble construction. The results presented in this paper, as well as in Appendix C.4, demonstrate that our method outperforms the state-of-the-art approach for ensemble construction, DE (Lakshminarayanan et al., 2017), on a range of benchmarks in OOD detection, while preserving the accuracy (Table C3).

Table 1: Averaged metrics across 10 OOD datasets.

| Model | Dataset | Deep Ensembles | | | Ours | | |
|---|---|---|---|---|---|---|---|
| | | AUC (↑) | AP (↑) | FPR95 (↓) | AUC (↑) | AP (↑) | FPR95 (↓) |
| PreResNet164 | C10 | 0.94 | 0.92 | 0.17 | **0.95** | **0.95** | **0.14** |
| | C100 | 0.79 | 0.80 | 0.47 | **0.88** | **0.88** | **0.40** |
| | SVHN | 0.99 | 0.97 | 0.02 | **1.00** | **0.98** | **0.01** |
| WideResNet28x10 | C10 | 0.95 | 0.94 | 0.15 | **0.96** | **0.96** | **0.12** |
| | C100 | 0.86 | 0.85 | 0.36 | **0.90** | **0.91** | **0.30** |
| | SVHN | 0.99 | 0.96 | 0.03 | **1.00** | **0.99** | **0.01** |
| VGG16BN | C10 | 0.92 | 0.91 | 0.23 | **0.95** | **0.95** | **0.18** |
| | C100 | 0.83 | 0.82 | 0.45 | **0.89** | **0.90** | **0.36** |
| | SVHN | 0.99 | 0.96 | 0.02 | **1.00** | **0.98** | 0.02 |

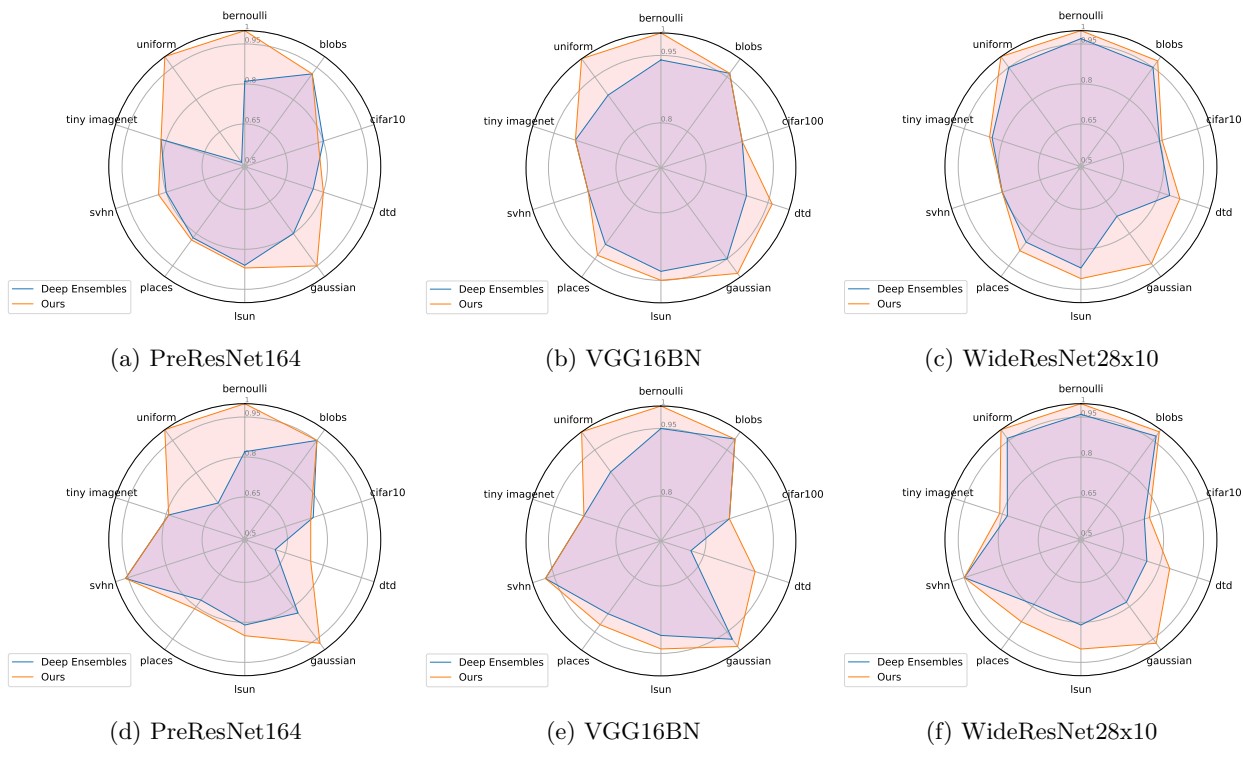

(a) PreResNet164      (b) VGG16BN      (c) WideResNet28x10

(d) PreResNet164      (e) VGG16BN      (f) WideResNet28x10

Figure 3: Out-of distribution detection results on CIFAR 100 for 3 different architectures (read column-wise). Here, we show AUC (top row) and AP values (bottom row) from 0.5 to 1 averaged across 5 seeds.

This study has some limitations, which outline several directions for the future work. Firstly, we did not compare our approach to a variety of existing methods for ensemble generation, e.g. snapshot ensembles (Huang et al., 2017), batch ensembles (Wen et al., 2020) or hyperparameter ensembles (Wenzel et al., 2020b). However, these methods are heuristic, and as discussed in the related work, our method can be used in conjunction with them to make them more principled. Furthermore, we note that the main contribution of this paper is novel theory.

The second limitation of this work is that it does not compare to f-POVI Wang et al. (2019); D'Angelo & Fortuin (2021). However, as noted earlier, those methods are in a different class, as they focus on training models without state-of-the-art training techniques such as batch normalization (Ioffe & Szegedy, 2015) and data augmentation.

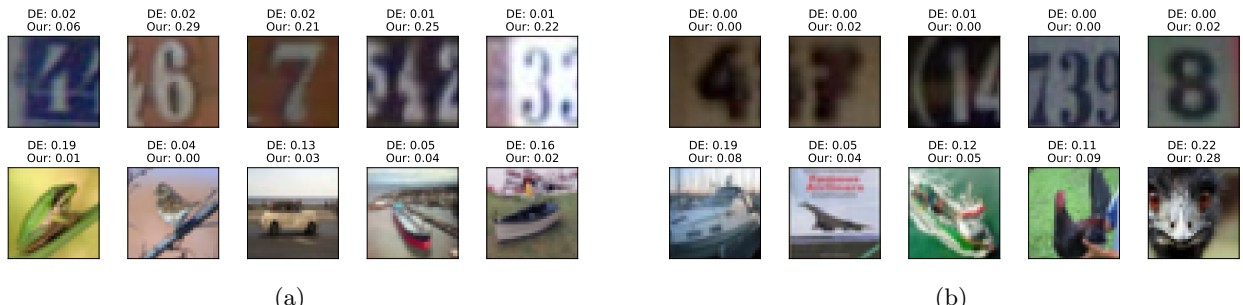

(a)                  (b)

Figure 4: OOD detection examples. In subplot (a), the top row shows true positives, and the bottom – true negatives detected by our method. We also show the uncertainty values. Subplot (b) shows the failures of both DE and our method on positives (top row) and negatives (bottom row), respectively. Here, we used PreResNet164 trained on CIFAR10 ($M = 11$). SVHN was used as an OOD dataset.

The third limitation is that we did not fully explore different techniques of generating weighting distributions for the diversity term. However, we still evaluated the possibility of using real images for this purpose in Appendix C.4, similarly to an outlier exposure setting (Hendrycks et al., 2019).

The fourth limitation of this work, is that we have implemented Algorithm 1 only for a single instance of an $f$-divergence with a single type of kernel density estimator, and Algorithm 2 is specific to neural networks. All this can alter the approximation factor on the $f$-divergence upper-bound, and it is thus hard to quantify the exact approximation gurarantees for the case of neural network posteriors. We consider that one can get results of this type only empirically, which is computationally intractable in general, but could still be evaluated on small scale problems in the future studies.

The final limitation is related to our method, as it lacks parallelization possibilities compared to DE. We note, however, that there is a class of parallel submodular optimization algorithms that enable parallelization and retain approximation guarantees (Ene & Nguyen, 2020). When both our method and DE are run sequentially on GPU, one can observe 70-90% computational overhead compared to DE when running our method for $M = 11$ (Table C9). Such an overhead, is, however, natural due to saving predictions and a double number of forward passes. We note here, that despite that training of the proposed method was roughly 1.75-1.9 times more expensive than DE, both methods had the same test runtime. Therefore, our approach is still of interest in applications where high quality uncertainty estimation at test time cannot be traded in favor of lower cost training.

To conclude, this paper provides a novel foundational framework for Bayesian Deep Learning. We hope for the wide adaption of the proposed method by practitioners across the fields. Our code is available at https://github.com/Oulu-IMEDS/greedy_ensembles_training.

### Acknowledgements

We acknowledge support from the Research Foundation - Flanders (FWO) through project numbers G0A1319N and S001421N, KU Leuven Internal Funds via the MACCHINA project, and funding from the Flemish Government under the "Onderzoeksprogramma Artificiële Intelligentie (AI) Vlaanderen" programme.

This research was also supported by the strategic funds of the University of Oulu, Finland, funding from the Academy of Finland (Profi6 336449 funding program and Finnish Center for Artificial Intelligence flagship), as well as the Northern Ostrobothnia hospital district, Finland (VTR project K33754). A.T. acknowledges travel support from the European Union's Horizon 2020 research and innovation programme under grant agreement No 951847. We thank CSC – Finnish Center for Science for generous computational resources. We also acknowledge the computational resources provided by the Aalto Science-IT project.

Iaroslav Melekhov, Markus Heinonen, Martin Trapp, Bas Veeling, and Egor Panfilov are acknowledged for useful suggestions that helped to improve this work.

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

# A  Proofs

## A.1  Proof of Theorem 1

To prove Theorem 1, we make use of the following theorem:

**Theorem A1** (Theorem 1.4 from (Mitrinovic & Vasic, 1970)). *A function $f : D \to R$ is convex on $D = [a, b]$ if and only if $\forall x_1 < x_2 < x_3 \in D$*

$$\begin{vmatrix} x_1 & f(x_1) & 1 \\ x_2 & f(x_2) & 1 \\ x_3 & f(x_3) & 1 \end{vmatrix} \geq 0. \tag{17}$$

**Theorem 1.** *Any $f$-divergence*

$$D_f(p||q_M) = \int f\left(\frac{p(z)}{\frac{1}{M}\sum_{j=1}^{M} K_j(z)}\right) \frac{1}{M}\sum_{m=1}^{M} K_m(z)dz \tag{18}$$

*between a distribution $p(z)$ and a mixture of $M$ kernels with equal weights is supermodualar in a cardinality-fixed setting, assuming that $\forall z \max_{q_M} D_f(p(z)||q_M(z)) < \infty$.*

*Proof.* In order to prove that $f$-divergences are supermodular, we need to show that $\forall \alpha > 0$, $xf(\frac{\alpha}{x})$ is convex, because $q_M(z)$ is a positive modular function (Bach, 2019, Proposition 6.1), due to $M$ being fixed.

From Theorem A1, $f(x)$ is convex on a closed interval $[a, b]$, *if and only if $\forall x_1 < x_2 < x_3$ in $[a, b]$*

$$\begin{vmatrix} x_1 & f(x_1) & 1 \\ x_2 & f(x_2) & 1 \\ x_3 & f(x_3) & 1 \end{vmatrix} \geq 0. \tag{19}$$

We know that $x_1 \leq x_2 \leq x_3$ and $x_1, \alpha > 0$. We divide each $i^{th}$ row by $x_i$:

$$\begin{vmatrix} 1 & \frac{1}{x_1}f(x_1) & \frac{1}{x_1} \\ 1 & \frac{1}{x_2}f(x_2) & \frac{1}{x_2} \\ 1 & \frac{1}{x_3}f(x_3) & \frac{1}{x_3} \end{vmatrix} \geq 0. \tag{20}$$

After the division, we denote new variables $y_1 = \frac{1}{x_3}$, $y_2 = \frac{1}{x_2}$, and $y_3 = \frac{1}{x_2}$. One can see that $y_1 < y_2 < y_3$, because $x_1 < x_2 < x_3$. We then get

$$\begin{vmatrix} 1 & y_3 f(\frac{1}{y_3}) & y_3 \\ 1 & y_2 f(\frac{1}{y_2}) & y_2 \\ 1 & y_1 f(\frac{1}{y_1}) & y_1 \end{vmatrix} \geq 0. \tag{21}$$

Changing the first and the third row of the determinant will change the sign. Changing the third and the first columns will also change the sign. Therefore

$$\begin{vmatrix} y_1 & y_1 f(\frac{1}{y_1}) & 1 \\ y_2 & y_2 f(\frac{1}{y_2}) & 1 \\ y_3 & y_3 f(\frac{1}{y_3}) & 1 \end{vmatrix} \geq 0, \tag{22}$$

and thus we get that

$$\forall x \in [a, b],\ xf\left(\frac{1}{x}\right) \text{ is convex} \iff f(x) \text{ is convex.} \tag{23}$$

Consider the following reparameterization $\tilde{y} = \frac{y}{\alpha}$, which preserves convexity. Then

$$
\begin{vmatrix}
\alpha \tilde{y}_1 & \alpha \tilde{y}_1 f(\frac{\alpha}{\tilde{y}_1}) & 1 \\
\alpha \tilde{y}_2 & \alpha \tilde{y}_2 f(\frac{\alpha}{\tilde{y}_2}) & 1 \\
\alpha \tilde{y}_3 & \alpha \tilde{y}_3 f(\frac{\alpha}{\tilde{y}_3}) & 1
\end{vmatrix} \geq 0.
\tag{24}
$$

Division of the first and the second column by $\alpha$ does not change the sign of the determinant, therefore,

$$
x f\left(\frac{\alpha}{x}\right) \text{ is convex} \iff f(x) \text{ is convex},
\tag{25}
$$

which concludes the proof. $\qquad\square$

## A.2 Proof of Proposition 1

**Proposition 1.** *Consider $C = \max D_f(p\|q_M)$, where $D_f(p\|q_M)$ is an arbitrary $f$-divergence between some distribution $p(z)$ and a mixture of kernels $q_M(z) = \frac{1}{M}\sum_{j=1}^{M} K_j(z)$, and $D_f(p\|q_M) < \infty$. Then, maximization of the marginal gain for the set function*

$$
F(Z) = -\int f\left(\frac{p(z)}{\frac{1}{M}\sum_{j=1}^{M} K_j(z)}\right) \frac{1}{M}\sum_{m=1}^{M} K_m(z) dz + C,
\tag{26}
$$

*at step $k$ of a greedy algorithm corresponds to*

$$
\arg\max_{z_k} \Delta(z_k|Z) = \arg\min_{z_k} \mathbb{E}_{z \sim K_k(z)} f\left(\frac{p(z)}{\frac{1}{M}\sum_{j=1}^{k} K_j(z)}\right).
\tag{27}
$$

*Proof.* We aim to derive a marginal gain of adding an element defined by $z_k$ to $\frac{1}{M}\sum_{j=1}^{k-1} K_j(z)$[1]. Let us denote $G(z) = f\left(\frac{Mp(z)}{\sum_{j=1}^{k-1} K_j(z)}\right) - f\left(\frac{Mp(z)}{\sum_{j=1}^{k} K_j(z)}\right)$. Then

$$
-\int f\left(\frac{Mp(z)}{\sum_{j=1}^{k} K_j(z)}\right) \frac{1}{M}\sum_{m=1}^{k} K_m(z) dz + C + \int f\left(\frac{Mp(z)}{\sum_{j=1}^{k-1} K_j(z)}\right) \frac{1}{M}\sum_{m=1}^{k-1} K_m(z) dz - C =
$$

$$
\int \frac{G(z)}{M}\sum_{m=1}^{k-1} K_m(z) dz - \int f\left(\frac{Mp(z)}{\sum_{j=1}^{k} K_j(z)}\right) \frac{K_k(z)}{M} dz.
\tag{28}
$$

One can observe that the first term of (28) is upper-bounded by a constant, which is not dependent on $z_k$:

$$
\int \frac{G(z)}{M}\sum_{m=1}^{k-1} K_m(z) dz \leq \int f\left(\frac{kp(z)}{\sum_{j=1}^{k-1} K_j(z)}\right) \frac{1}{M}\sum_{j=1}^{k-1} K_j(z) dz = \text{const},
\tag{29}
$$

therefore, to maximize the marginal gain, one needs to maximize the second term. Consequently, we write the objective corresponding to a marginal gain as

$$
\Delta(z_k | Z \setminus z_k) = -\int f\left(\frac{p(z)}{\frac{1}{M}\sum_{j=1}^{k} K_j(z)}\right) K_k(z) dz,
\tag{30}
$$

maximization of which is equivalent to

$$
\arg\min_{z_k} \mathbb{E}_{z \sim K_k(z)} f\left(\frac{p(z)}{\frac{1}{M}\sum_{j=1}^{k} K_j(z)}\right),
\tag{31}
$$

which concludes the proof. $\qquad\square$

---

[1]Note: $\frac{1}{M}$ is a constant, which remains unchanged at all iterations of the greedy algorithm.

# B  Implementation details

## B.1  Weighting distribution for the diversity term

We propose the following simple heuristic, defining $p^*(x)$ as a normal distribution $\mathcal{N}(\mu_\mathcal{D}, \alpha \cdot \Sigma_\mathcal{D})$ of dimensionality, corresponding to the training data. The covariance $\Sigma_\mathcal{D}$ for this distribution is set to be diagonal, such that the variance for every dimension $j$ is $\Sigma_\mathcal{D}[j,j] = (\alpha \cdot \sigma_j)^2$, where $\alpha > 1$ is a scaling parameter, and $\sigma_j^2$ is a variance of the dimension $j$ computed from samples of the training dataset $\mathcal{D}$. Similarly, $\mu_\mathcal{D}$, the vector of expected values for every dimension, is also computed from the training data. Finally, the hyperparameter $\alpha = 5$ was found to work well, and we thus report all the experimental results with it fixed. We note that a similar technique, but for *in-distribution* data generation, has been used earlier in (Melville & Mooney, 2005).

## B.2  OOD detection benchmarks

**MNIST**  The MNIST dataset benchmark included 6 datasets: Fashion MNIST Xiao et al. (2017), DTD Cimpoi et al. (2014), Omniglot Lake et al. (2015), Gaussian noise, Bernoulli noise, and uniform noise datasets (see Table B1). We re-scaled all the images to the range of $[0, 1]$. Subsequently, we applied the same mean and standard deviation as we have applied to the original images before feeding them to the network.

**CIFAR and SVHN**  The CIFAR and SVHN OOD benchmark included 10 different datasets. Here, we also DTD, Bernoulli noise, Gaussian noise and uniform noise datasets, and added Places 365 (Zhou et al., 2017), Tiny ImageNet (Le & Yang, 2015; Deng et al., 2009), and LSUN (Yu et al., 2015) datasets to the benchmark. For CIFAR10 as in-domain data, we added CIFAR100 and SVHN (Netzer et al., 2011) to the benchmark. For CIFAR100 – CIFAR10 and SVHN. Finally, for SVHN, we added CIFAR10 and CIFAR100 as OOD datasets, making a total of 10 OOD datasets per 1 in-distribution dataset. The details about each of the datasets are shown in Table B1.

Before feeding the images to the network, we applied re-scaling similarly to the MNIST setting. We also used the same mean and standard deviation normalization procedure, as for the in-domain data.

Table B1: Description of the datasets used in all the exp. R indicates real images, S – synthetic.

| Dataset | Type | # samples | Comment |
|---|---|---|---|
| Uniform | | $25,000$ | N/A |
| Gaussian | | $25,000$ | Generated once, used in all experiments |
| Blobs | S | $25,000$ | N/A |
| Bernoulli | | $25,000$ | N/A |
| Omniglot | | $13,181$ | Evaluation images |
| CIFAR10 | | $10,000$ | Test set (not used in training) |
| CIFAR100 | | $10,000$ | Test set (not used in training) |
| SVHN | | $73,257$ | Test set (not used in training) |
| Places 365 | | $10,000$ | First $10,000$ images from the test set (sorted alphabetically) |
| TinyImageNet | R | $10,000$ | Original validation set images |
| DTD | | $5,640$ | Release 1.0.1 |
| LSUN | | $10,000$ | Test set |
| Fashion MNIST | | $10,000$ | Test set |

## B.3  Epistemic uncertainty computation

We used the *epistemic uncertainty*, i.e. mutual information (MI) between the label $y$ for the point $\hat{\mathbf{x}}$ and the posterior distribution over functions $p(f|\mathcal{D})$, to evaluate the uncertainty (Malinin & Gales, 2018; Depeweg et al., 2018). As a distribution over weights induces a distribution over functions, we approximate the MI as:

$$\mathcal{I}(y; f|\hat{\mathbf{x}}, \mathcal{D}) = \mathcal{H}\left[\mathbb{E}_{p(\theta|\mathcal{D})} p(y \mid \theta, \hat{\mathbf{x}}, \mathcal{D})\right] - \mathbb{E}_{p(\theta|\mathcal{D})} \mathcal{H}\left[p(y \mid \theta, \hat{\mathbf{x}}, \mathcal{D})\right], \tag{32}$$

where $\mathcal{H}[\cdot]$ denotes the entropy. One can see that this metric can be efficiently computed from the predictions of an ensemble.

## C  Experiments

### C.1  Experimental details

**Model selection**  Contrary to the commonly used practice, we did not use CIFAR10/100 and SVHN test set sets for model selection. Neither did we use any OOD data. Instead, we used validation set accuracy (10% of the training data; randomly chosen stratified split) to select the models when optimizing the marginal gain. The best snapshot was found using the validation data, was then selected for final testing. When selecting the models for evaluation on OOD data, we first evaluated ensembles on the in-distribution test set (Appendix C.2). Subsequently, we selected the highest $\lambda_M$ that did not harm the test set (in-domain) performance (no overlap of confidence intervals defined as mean $\pm$ standard error). To provide additional information, we also analyzed adaptive calibration error (ACE) with 30 bins (Nixon et al., 2019).

**Two moons dataset**  For the synthetic data experiments, we used scikit-learn (Pedregosa et al., 2011), and generated a two-moons dataset with 300 points in total, having the noise parameter fixed to 0.3. Here, we used a two-layer neural network with ReLU (Krizhevsky, 2009) activations and hidden layer size of 128.

**CIFAR10/100 and SVHN**  The main training hyper-parameters were adapted from (Maddox et al., 2019) (see Table C2), but with additional modifications inspired by (Malinin & Gales, 2018; Smith & Topin, 2019), which helped to train the CIFAR models to state-of-the-art performance in only 100 epochs. As such, we first employed a warm-up of the learning rate ($LR$) from a value 10 times lower than the initial $LR$ ($LR_{init}$ in Table C2) for 5 epochs. Subsequently, after 50% of the training budget, we linearly annealed the $LR$ to the value of $LR \times lr_{scale}$ until 90% of the training budget is reached, after which we kept the value of $LR$ constant.

All models were trained using stochastic gradient descent with momentum of 0.9 and a total batch size of 128. We employed standard training augmentations – horizontal flipping, reflective padding to $34 \times 34$, and random crop to $34 \times 34$ pixels.

| Model | $LR_{init}$ | Nesterov | Weight Decay | $lr_{scale}$ |
|---|---|---|---|---|
| PreResNet164 | 0.1 | Yes | 0.0001 | 0.01 |
| VGG16BN | 0.05 | No | 0.0005 | 0.01 |
| WideResNet28x10 | 0.1 | No | 0.0005 | 0.001 |

Table C2: Main hyper-parameters of all the models used in the CIFAR and SVHN in-domain experiments.

**MNIST**  In addition to the CIFAR10/100 experiments, we also trained our method on MNIST (LeCun et al., 1998) with PreResNet8 architecture. As OOD, we used FashionMNIST (Xiao et al., 2017) and Omniglot (Lake et al., 2015) datasets. We also tested other architectures, such as PreResNet20, but the models with higher depth than 8 already gave nearly perfect scores on MNIST.

Hyper-parameter-wise, we trained all the models for 20 epochs without warmup with the batch size of 256 using plain SGD with momentum. The weight decay was set to $1e - 5$. We used $LR$ annealing similarly as for CIFAR experiments, but used $lr_{scale} = 0.0001$. No data augmentations were used in any of the MNIST experiments. $\lambda_M$ was searched in range $\{0.0001, 0.001, 0.01, 0.11, 7\}$ for $M \in \{3, 5, 9, 15\}$. This series of experiments was re-run 3 times, as the MNIST dataset is rather simple, and the test scores have low variance between the runs.

### C.2  CIFAR10/100 and SVHN in-domain performance

**CIFAR10/100 in-distribution performance vs. diversity.**  Figure C1 provides an illustration of how the test set performance changes with $\lambda_M$ on CIFAR data. One can see a general trend that when $\lambda_M$

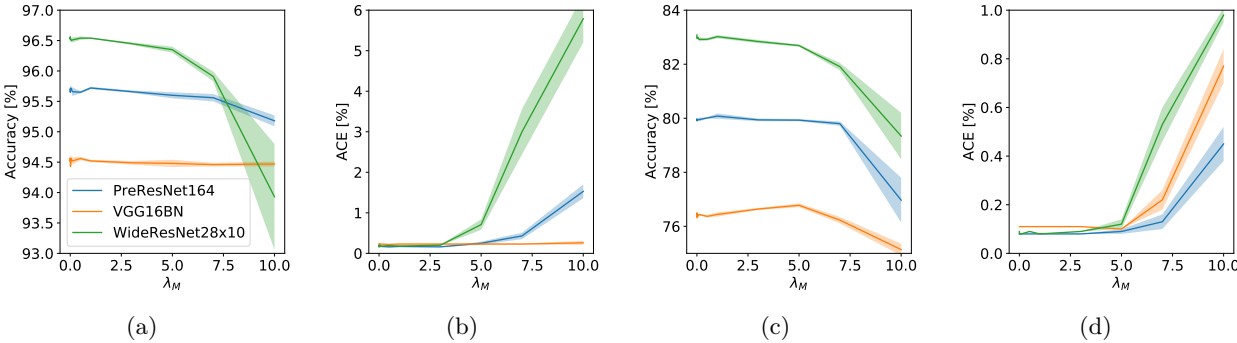

Figure C1: Relationship between accuracy, ACE, and $\lambda_M$ ($M = 11$). Subplots (a) and (b) show the results for CIFAR10. Subplots (c) and (d) show the results for CIFAR100.

approaches $M$, the models lose the ability to make accurate predictions, which results in lower accuracy and poorer calibration. Interestingly, performance on the VGG model degrades much slower with $\lambda_M$ compared to other architectures. Similar findings were also obtained for the SVHN dataset. Based on the test performance, we selected the models for further evaluation on OOD benchmark.

**CIFAR and SVHN: best models' performance**   Table C3 shows the results of all the trained models on the in-domain data. One can see that the results between Deep Ensembles (DE) (Lakshminarayanan et al., 2017) do not differ significantly. We trained all these models according to the earlier specified hyper-parameters and the learning rate schedule. Models selected in Table C3 are used to report the results in the main experiments.

Table C3: In-domain performance on the test sets of CIFAR10/100 and SVHN for all the models used in the experiments ($M = 11$). We report mean and standard error over 5 random seeds for each of the models. Standard errors are reported if they are more than 0.01 across runs. DE indicates Deep Ensembles.

| Architecture | Dataset | Method | Accuracy (%) | NLL $\times 100$ | ACE (%) |
|---|---|---|---|---|---|
| PreResNet164 | C10 | DE | $95.70_{\pm 0.02}$ | $13.28_{\pm 0.06}$ | $0.18$ |
| | | Ours ($\lambda_M = 3$) | $95.66_{\pm 0.02}$ | $13.18_{\pm 0.09}$ | $0.16$ |
| | C100 | DE | $79.97_{\pm 0.04}$ | $73.44_{\pm 0.17}$ | $0.08$ |
| | | Ours ($\lambda_M = 5$) | $79.93_{\pm 0.04}$ | $74.16_{\pm 0.91}$ | $0.09_{\pm 0.01}$ |
| | SVHN | DE | $99.46_{\pm 0.01}$ | $2.34_{\pm 0.04}$ | $0.25$ |
| | | Ours ($\lambda_M = 1$) | $99.38_{\pm 0.01}$ | $3.16_{\pm 0.13}$ | $0.81_{\pm 0.12}$ |
| VGG16BN | C10 | DE | $94.55_{\pm 0.02}$ | $17.59_{\pm 0.05}$ | $0.23$ |
| | | Ours ($\lambda_M = 5$) | $94.48_{\pm 0.06}$ | $17.84_{\pm 0.15}$ | $0.23_{\pm 0.01}$ |
| | C100 | DE | $76.32_{\pm 0.09}$ | $91.07_{\pm 0.35}$ | $0.11$ |
| | | Ours ($\lambda_M = 5$) | $76.78_{\pm 0.07}$ | $89.10_{\pm 0.33}$ | $0.10$ |
| | SVHN | DE | $99.40_{\pm 0.01}$ | $2.71_{\pm 0.02}$ | $0.18_{\pm 0.01}$ |
| | | Ours ($\lambda_M = 1$) | $99.25_{\pm 0.01}$ | $3.94_{\pm 0.10}$ | $0.95_{\pm 0.12}$ |
| WideResNet28x10 | C10 | DE | $96.56_{\pm 0.02}$ | $10.76_{\pm 0.06}$ | $0.16_{\pm 0.01}$ |
| | | Ours ($\lambda_M = 1$) | $96.54_{\pm 0.01}$ | $10.99_{\pm 0.03}$ | $0.18_{\pm 0.01}$ |
| | C100 | DE | $83.08_{\pm 0.09}$ | $62.05_{\pm 0.18}$ | $0.09$ |
| | | Ours ($\lambda_M = 1$) | $83.02_{\pm 0.06}$ | $62.20_{\pm 0.13}$ | $0.08$ |
| | SVHN | DE | $99.45_{\pm 0.01}$ | $2.53_{\pm 0.03}$ | $0.43_{\pm 0.01}$ |
| | | Ours ($\lambda_M = 1$) | $99.38$ | $2.87_{\pm 0.04}$ | $0.46_{\pm 0.01}$ |

## C.3 Detailed CIFAR, SVHN results

Detalized versions of the results presented in the main text are shown in Table C4. The corresponding $\lambda_M$ coefficients are the same as in Table C3.

Table C4: CIFAR10 results. We report mean and standard error over 5 random seeds for each of the models. Standard errors are reported if they are more than 0.01 across runs. DE indicates Deep Ensembles.

| Architecture | OOD dataset | DE | | | Ours | | |
|---|---|---|---|---|---|---|---|
| | | AUC ($\uparrow$) | AP ($\uparrow$) | FPR95 ($\downarrow$) | AUC ($\uparrow$) | AP ($\uparrow$) | FPR95 ($\downarrow$) |
| PreResNet164 | bernoulli | $0.98_{\pm0.01}$ | $0.97_{\pm0.01}$ | $0.04_{\pm0.01}$ | **1.00** | **1.00** | **0.00** |
| | blobs | 0.96 | 0.98 | $0.12_{\pm0.01}$ | 0.96 | 0.98 | $0.12_{\pm0.01}$ |
| | cifar100 | 0.90 | 0.87 | 0.30 | 0.90 | **0.88** | 0.30 |
| | dtd | 0.93 | 0.83 | $0.19_{\pm0.01}$ | **0.96** | **0.93** | **0.14** |
| | gaussian | $0.93_{\pm0.01}$ | $0.94_{\pm0.01}$ | $0.16_{\pm0.01}$ | $0.96_{\pm0.02}$ | $0.97_{\pm0.02}$ | $\mathbf{0.11_{\pm0.03}}$ |
| | lsun | 0.93 | 0.89 | 0.20 | **0.95** | **0.94** | **0.18** |
| | places | 0.92 | 0.89 | 0.21 | **0.94** | **0.93** | **0.19** |
| | svhn | 0.94 | 0.99 | 0.16 | **0.95** | 0.99 | **0.14** |
| | tiny imagenet | 0.91 | 0.88 | 0.28 | **0.92** | **0.89** | **0.26** |
| | uniform | $0.98_{\pm0.01}$ | $0.97_{\pm0.01}$ | $0.04_{\pm0.01}$ | **1.00** | **1.00** | **0.00** |
| VGG16BN | bernoulli | $0.94_{\pm0.01}$ | $0.95_{\pm0.01}$ | $0.11_{\pm0.02}$ | **1.00** | **1.00** | **0.00** |
| | blobs | 0.96 | 0.98 | 0.16 | 0.96 | 0.98 | **0.15** |
| | cifar100 | 0.89 | 0.86 | 0.34 | 0.89 | 0.86 | 0.34 |
| | dtd | 0.90 | $0.77_{\pm0.01}$ | $0.25_{\pm0.01}$ | **0.96** | **0.92** | **0.18** |
| | gaussian | 0.95 | 0.97 | $0.14_{\pm0.01}$ | **0.99** | **0.99** | $\mathbf{0.05_{\pm0.01}}$ |
| | lsun | 0.93 | 0.91 | 0.24 | **0.95** | **0.94** | **0.21** |
| | places | 0.91 | 0.90 | $0.27_{\pm0.01}$ | **0.94** | **0.93** | **0.23** |
| | svhn | 0.87 | 0.97 | $0.28_{\pm0.01}$ | 0.87 | 0.97 | $0.27_{\pm0.01}$ |
| | tiny imagenet | 0.90 | 0.88 | 0.33 | 0.90 | 0.88 | **0.32** |
| | uniform | $0.90_{\pm0.02}$ | $0.89_{\pm0.01}$ | $0.16_{\pm0.02}$ | **1.00** | **1.00** | **0.00** |
| WideResNet28x10 | bernoulli | 1.00 | 1.00 | 0.00 | 1.00 | 1.00 | 0.00 |
| | blobs | 0.96 | 0.97 | $0.11_{\pm0.01}$ | **0.97** | **0.98** | $0.10_{\pm0.01}$ |
| | cifar100 | **0.92** | 0.89 | 0.27 | 0.91 | 0.89 | 0.27 |
| | dtd | 0.93 | $0.86_{\pm0.01}$ | $0.22_{\pm0.01}$ | **0.97** | **0.94** | $\mathbf{0.14_{\pm0.01}}$ |
| | gaussian | $0.96_{\pm0.01}$ | $0.97_{\pm0.01}$ | $0.09_{\pm0.01}$ | **1.00** | **1.00** | **0.00** |
| | lsun | 0.93 | 0.91 | 0.20 | **0.95** | **0.95** | $\mathbf{0.17_{\pm0.01}}$ |
| | places | 0.93 | 0.91 | 0.21 | **0.95** | **0.94** | **0.18** |
| | svhn | **0.96** | 0.99 | **0.12** | 0.95 | 0.99 | $0.13_{\pm0.01}$ |
| | tiny imagenet | 0.92 | 0.90 | 0.27 | **0.93** | **0.91** | **0.25** |
| | uniform | 1.00 | 1.00 | 0.00 | 1.00 | 1.00 | 0.00 |

### C.3.1 MNIST detailed results

**In-domain performance**  Model selection scheme on MNIST was exactly the same as for the CIFAR10/100. We illustrate the relationship between $\lambda_M$, accuracy, and the Adaptive Calibration error (ACE) in Figure C2. One can see that $\lambda_M$ remains the same even when $M$ is increasing. Table C7 shows the detailed in-domain performance for the best $\lambda_M$, equal to 0.1.

**Out of distribution detection**  The final OOD benchmark results on MNIST are summarized in Table C8. One can see that on MNIST, our method yields a performance boost for any size of an ensemble, even for the very small ones ($M = 3$).

## C.4 Additional results

**Computational overhead**  As noted in the main section, our method has computation overhead compared to DE, which is also a parallel method, allowing to get the results computed in $\mathcal{O}(1)$ time (in the number of

Table C5: CIFAR100 results. We report mean and standard error over 5 random seeds for each of the models. Standard errors are reported if they are more than 0.01 across runs. DE indicates Deep Ensembles.

| Architecture | OOD dataset | DE | | | Ours | | |
|---|---|---|---|---|---|---|---|
| | | AUC ($\uparrow$) | AP ($\uparrow$) | FPR95 ($\downarrow$) | AUC ($\uparrow$) | AP ($\uparrow$) | FPR95 ($\downarrow$) |
| PreResNet164 | bernoulli | $0.81_{\pm0.03}$ | $0.82_{\pm0.03}$ | $0.24_{\pm0.04}$ | **1.00** | **1.00** | **0.00** |
| | blobs | $0.92_{\pm0.01}$ | 0.95 | $0.25_{\pm0.02}$ | $0.92_{\pm0.02}$ | $0.95_{\pm0.01}$ | $0.23_{\pm0.03}$ |
| | cifar10 | **0.80** | **0.76** | **0.57** | $0.78_{\pm0.01}$ | 0.75 | $0.67_{\pm0.02}$ |
| | dtd | $0.76_{\pm0.01}$ | $0.61_{\pm0.01}$ | $\mathbf{0.64_{\pm0.01}}$ | $\mathbf{0.80_{\pm0.01}}$ | $\mathbf{0.75_{\pm0.01}}$ | $0.78_{\pm0.05}$ |
| | gaussian | $0.80_{\pm0.01}$ | $0.83_{\pm0.01}$ | $0.37_{\pm0.02}$ | $\mathbf{0.95_{\pm0.02}}$ | $\mathbf{0.97_{\pm0.01}}$ | $\mathbf{0.16_{\pm0.05}}$ |
| | lsun | 0.86 | 0.81 | **0.45** | **0.87** | **0.85** | $0.47_{\pm0.01}$ |
| | places | 0.82 | 0.77 | **0.52** | **0.83** | **0.81** | $0.60_{\pm0.03}$ |
| | svhn | $0.80_{\pm0.01}$ | 0.96 | $0.55_{\pm0.01}$ | $\mathbf{0.83_{\pm0.01}}$ | 0.96 | $\mathbf{0.50_{\pm0.01}}$ |
| | tiny imagenet | 0.82 | 0.79 | **0.53** | 0.82 | 0.79 | $0.58_{\pm0.01}$ |
| | uniform | $0.51_{\pm0.09}$ | $0.66_{\pm0.05}$ | $0.55_{\pm0.09}$ | **1.00** | **1.00** | **0.00** |
| VGG16BN | bernoulli | $0.87_{\pm0.02}$ | $0.88_{\pm0.02}$ | $0.24_{\pm0.03}$ | **1.00** | **1.00** | **0.00** |
| | blobs | 0.95 | 0.97 | $0.16_{\pm0.01}$ | **0.97** | **0.98** | $\mathbf{0.12_{\pm0.02}}$ |
| | cifar10 | 0.78 | 0.73 | 0.63 | 0.78 | **0.74** | $0.64_{\pm0.01}$ |
| | dtd | 0.73 | 0.53 | $0.62_{\pm0.01}$ | **0.86** | $\mathbf{0.80_{\pm0.01}}$ | $\mathbf{0.50_{\pm0.01}}$ |
| | gaussian | $0.87_{\pm0.02}$ | $0.90_{\pm0.01}$ | $0.35_{\pm0.04}$ | $\mathbf{0.95_{\pm0.01}}$ | $\mathbf{0.97_{\pm0.01}}$ | $\mathbf{0.15_{\pm0.03}}$ |
| | lsun | 0.85 | 0.82 | 0.47 | **0.90** | **0.89** | **0.40** |
| | places | 0.82 | 0.78 | 0.55 | **0.86** | **0.85** | **0.51** |
| | svhn | $0.76_{\pm0.01}$ | 0.95 | $0.67_{\pm0.02}$ | $0.76_{\pm0.01}$ | 0.95 | $0.72_{\pm0.04}$ |
| | tiny imagenet | 0.81 | 0.78 | 0.56 | **0.83** | **0.79** | **0.55** |
| | uniform | $0.86_{\pm0.02}$ | $0.89_{\pm0.02}$ | $0.28_{\pm0.04}$ | **1.00** | **1.00** | **0.00** |
| WideResNet28x10 | bernoulli | $0.97_{\pm0.02}$ | $0.96_{\pm0.02}$ | $0.04_{\pm0.02}$ | **1.00** | **1.00** | **0.00** |
| | blobs | 0.95 | 0.97 | $0.16_{\pm0.01}$ | $\mathbf{0.98_{\pm0.01}}$ | **0.99** | $\mathbf{0.09_{\pm0.02}}$ |
| | cifar10 | 0.80 | 0.74 | 0.53 | **0.81** | **0.76** | $0.54_{\pm0.01}$ |
| | dtd | $0.84_{\pm0.01}$ | $0.75_{\pm0.01}$ | $0.54_{\pm0.01}$ | $\mathbf{0.88_{\pm0.01}}$ | $\mathbf{0.84_{\pm0.01}}$ | $\mathbf{0.49_{\pm0.02}}$ |
| | gaussian | $0.72_{\pm0.08}$ | $0.78_{\pm0.05}$ | $0.39_{\pm0.09}$ | $\mathbf{0.94_{\pm0.03}}$ | $\mathbf{0.97_{\pm0.01}}$ | $\mathbf{0.20_{\pm0.07}}$ |
| | lsun | 0.87 | $0.81_{\pm0.01}$ | 0.35 | **0.91** | $\mathbf{0.90_{\pm0.01}}$ | $\mathbf{0.32_{\pm0.01}}$ |
| | places | 0.84 | $0.79_{\pm0.01}$ | $0.44_{\pm0.01}$ | **0.88** | $\mathbf{0.87_{\pm0.01}}$ | $\mathbf{0.41_{\pm0.01}}$ |
| | svhn | 0.80 | 0.95 | $0.52_{\pm0.01}$ | $0.80_{\pm0.01}$ | 0.95 | $0.52_{\pm0.01}$ |
| | tiny imagenet | 0.84 | 0.78 | 0.46 | **0.85** | **0.81** | 0.46 |
| | uniform | $0.95_{\pm0.01}$ | $0.96_{\pm0.01}$ | $0.12_{\pm0.03}$ | **1.00** | **1.00** | **0.00** |

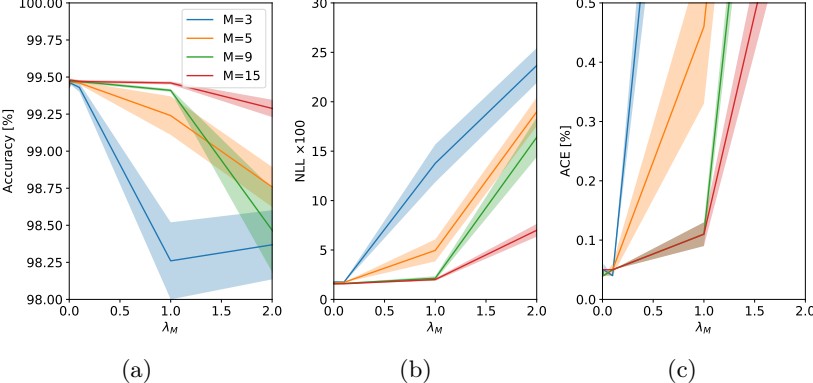

(a)  (b)  (c)

Figure C2: Relationship between accuracy, negative log-likelihood, ACE, and $\lambda_M$ for different $M$ on MNIST (LeCun et al., 1998). Subplots (a) and (b) show the results for PreResNet8. Experiments were re-run 3 times with different seeds. We found that the best results were obtained with $\lambda_M = 0.1$.

models). Table C9 shows the exact walltime (WT) comparisons and the estimated computational overhead computed as

$$Overhead = \left( \frac{WT_{Ours}}{WT_{DE}} - 1 \right) \times 100\%. \tag{33}$$

Table C6: SVHN results (averaged across 5 seeds)

| Architecture | OOD dataset | DE | | | Ours | | |
|---|---|---|---|---|---|---|---|
| | | AUC (↑) | AP (↑) | FPR95 (↓) | AUC (↑) | AP (↑) | FPR95 (↓) |
| PreResNet164 | bernoulli | 1.00 | 0.99 | 0.01 | 1.00 | **1.00** | **0.00** |
| | blobs | **1.00** | **0.99** | **0.01** | 0.99 | 0.98 | 0.02 |
| | cifar10 | 0.99 | 0.97 | 0.02 | 0.99 | 0.97 | 0.02 |
| | cifar100 | 0.99 | 0.96 | **0.02** | 0.99 | 0.96 | 0.04 |
| | dtd | 0.99 | 0.94 | 0.02 | **1.00** | **0.98** | **0.01** |
| | gaussian | 1.00 | 0.99 | 0.01 | 1.00 | **1.00** | **0.00** |
| | lsun | 0.99 | 0.96 | 0.02 | **1.00** | **0.98** | **0.01** |
| | places | 0.99 | 0.96 | 0.02 | **1.00** | **0.98** | **0.01** |
| | tiny imagenet | 0.99 | 0.96 | 0.02 | **1.00** | **0.97** | 0.02 |
| | uniform | 1.00 | 0.99 | 0.01 | 1.00 | **1.00** | **0.00** |
| VGG16BN | bernoulli | 1.00 | 0.99 | 0.01 | 1.00 | **1.00** | **0.00** |
| | blobs | 1.00 | 0.98 | 0.01 | 1.00 | **0.99** | 0.01 |
| | cifar10 | 0.99 | 0.95 | **0.02** | 0.99 | **0.96** | 0.03 |
| | cifar100 | 0.99 | 0.94 | **0.02** | 0.99 | **0.95** | 0.05 |
| | dtd | 0.99 | 0.93 | 0.02 | **1.00** | **0.97** | **0.01** |
| | gaussian | 1.00 | 0.99 | 0.01 | 1.00 | **1.00** | **0.00** |
| | lsun | 0.99 | 0.95 | 0.02 | **1.00** | **0.99** | **0.01** |
| | places | 0.99 | 0.96 | 0.02 | **1.00** | **0.98** | **0.01** |
| | tiny imagenet | 0.99 | 0.95 | **0.02** | 0.99 | **0.97** | 0.03 |
| | uniform | 1.00 | 0.99 | 0.01 | 1.00 | **1.00** | **0.00** |
| WideResNet28x10 | bernoulli | 1.00 | $0.99_{\pm 0.01}$ | $0.02_{\pm 0.01}$ | 1.00 | 1.00 | **0.00** |
| | blobs | 0.99 | 0.98 | 0.02 | **1.00** | **0.99** | **0.01** |
| | cifar10 | 0.99 | 0.96 | 0.02 | **1.00** | **0.97** | 0.02 |
| | cifar100 | 0.99 | 0.95 | 0.03 | 0.99 | **0.97** | **0.02** |
| | dtd | 0.99 | $0.91_{\pm 0.01}$ | 0.04 | **1.00** | **0.99** | **0.00** |
| | gaussian | 1.00 | 0.98 | 0.02 | 1.00 | **1.00** | **0.00** |
| | lsun | 0.99 | 0.95 | 0.03 | **1.00** | **0.99** | **0.00** |
| | places | 0.99 | 0.95 | 0.03 | **1.00** | **0.99** | **0.00** |
| | tiny imagenet | 0.99 | 0.96 | 0.02 | **1.00** | **0.98** | **0.01** |
| | uniform | 0.99 | $0.98_{\pm 0.01}$ | $0.03_{\pm 0.01}$ | **1.00** | **1.00** | **0.00** |

| Size | Method | Accuracy (%) | NLL ×100 | ECE (%) |
|---|---|---|---|---|
| 3 | DE | $99.44_{\pm 0.02}$ | $1.72_{\pm 0.03}$ | $0.23_{\pm 0.01}$ |
| | Ours | $99.43_{\pm 0.03}$ | $1.74_{\pm 0.07}$ | $0.24_{\pm 0.04}$ |
| 5 | DE | $99.43_{\pm 0.01}$ | $1.65_{\pm 0.04}$ | $0.22_{\pm 0.01}$ |
| | Ours | $99.46_{\pm 0.01}$ | $1.72_{\pm 0.02}$ | $0.32_{\pm 0.01}$ |
| 7 | DE | $99.44_{\pm 0.01}$ | $1.61_{\pm 0.03}$ | $0.25_{\pm 0.01}$ |
| | Ours | $99.41_{\pm 0.01}$ | $1.66_{\pm 0.06}$ | 0.28 |
| 9 | DE | $99.44_{\pm 0.01}$ | $1.65_{\pm 0.04}$ | $0.28_{\pm 0.01}$ |
| | Ours | $99.47_{\pm 0.01}$ | $1.62_{\pm 0.03}$ | $0.31_{\pm 0.01}$ |
| 15 | DE | $99.47_{\pm 0.01}$ | $1.60_{\pm 0.03}$ | $0.29_{\pm 0.01}$ |
| | Ours | 99.49 | $1.57_{\pm 0.03}$ | $0.30_{\pm 0.02}$ |

Table C7: Test set results (in-domain performance) of our method on PreResNet8 trained on MNIST with different ensemble sizes $M$. We report the means over 3 random seeds for each of the models. $\lambda_M = 0.1$ Was used for all the experiments when training our method. Standard errors are reported if they are more than 0.01 across runs. DE indicates Deep Ensembles.

**Does the choice weighting distribution for the diversity term affect the results?** We considered PreResNet164 trained on CIFAR100 with CIFAR10 as a weighting distribution dataset. This approach can be seen as a form of outlier exposure technique, earlier proposed by Hendrycks et al. (2019). The results on OOD benchmark (excluding CIFAR10) are shown in Table C10. One can observe that the relationship between changing to a dataset of real images and the OOD detection performance is non-trivial. While for some datasets outlier exposure did bring benefit, for some other datasets, it did not. Notably, the datasets on

Table C8: Out-of-distribution detection results of our method with PreResNet8 trained on MNIST with different ensemble sizes $M$. We report the means over 3 random seeds for each of the models. $\lambda_M = 0.1$ Was used for all the experiments when training our method. Standard errors are reported if they are more than 0.01 across runs. DE indicates Deep Ensembles.

| Size | OOD dataset | DE | | | Ours | | |
|------|-------------|---------|--------|-----------|---------|--------|-----------|
| | | AUC (↑) | AP (↑) | FPR95 (↓) | AUC (↑) | AP (↑) | FPR95 (↓) |
| 3 | bernoulli | $0.09_{\pm0.05}$ | $0.52_{\pm0.01}$ | $0.98$ | **1.00** | **1.00** | **0.00** |
| | dtd | $0.41_{\pm0.16}$ | $0.47_{\pm0.13}$ | $0.97_{\pm0.01}$ | **1.00** | **0.99** | **0.00** |
| | fashion mnist | $0.89_{\pm0.03}$ | $0.92_{\pm0.02}$ | $0.71_{\pm0.19}$ | **$0.99_{\pm0.01}$** | **0.99** | **$0.07_{\pm0.03}$** |
| | gaussian | $0.98_{\pm0.01}$ | $0.99_{\pm0.01}$ | $0.06_{\pm0.02}$ | $0.99_{\pm0.01}$ | 1.00 | $0.03_{\pm0.02}$ |
| | omniglot | $0.98$ | $0.98$ | $0.08$ | 0.98 | 0.98 | 0.08 |
| | uniform | $0.23_{\pm0.17}$ | $0.58_{\pm0.07}$ | $0.90_{\pm0.06}$ | **1.00** | **1.00** | **0.00** |
| 5 | bernoulli | $0.02$ | $0.50$ | $0.98$ | **1.00** | **1.00** | **0.00** |
| | dtd | $0.48_{\pm0.17}$ | $0.55_{\pm0.14}$ | $0.94_{\pm0.03}$ | **1.00** | **1.00** | **0.00** |
| | fashion mnist | $0.87_{\pm0.05}$ | $0.92_{\pm0.03}$ | $0.67_{\pm0.24}$ | **0.99** | **0.99** | **$0.04_{\pm0.01}$** |
| | gaussian | $1.00$ | $1.00$ | $0.02_{\pm0.01}$ | 1.00 | 1.00 | $0.01_{\pm0.01}$ |
| | omniglot | $0.98$ | $0.99$ | **0.06** | 0.98 | 0.99 | 0.07 |
| | uniform | $0.30_{\pm0.22}$ | $0.62_{\pm0.10}$ | $0.77_{\pm0.18}$ | **1.00** | **1.00** | **0.00** |
| 7 | bernoulli | $0.33_{\pm0.24}$ | $0.66_{\pm0.13}$ | $0.77_{\pm0.18}$ | **1.00** | **1.00** | **0.00** |
| | dtd | $0.75_{\pm0.12}$ | $0.76_{\pm0.11}$ | $0.67_{\pm0.25}$ | **1.00** | **1.00** | **0.00** |
| | fashion mnist | $0.95_{\pm0.02}$ | $0.97_{\pm0.02}$ | $0.29_{\pm0.18}$ | **0.99** | 0.99 | **$0.05_{\pm0.01}$** |
| | gaussian | $1.00$ | $1.00$ | $0.02_{\pm0.01}$ | 1.00 | 1.00 | $0.02_{\pm0.01}$ |
| | omniglot | $0.99$ | $0.99$ | $0.06$ | 0.99 | 0.99 | $0.06_{\pm0.01}$ |
| | uniform | $0.49_{\pm0.22}$ | $0.71_{\pm0.12}$ | $0.64_{\pm0.26}$ | **1.00** | **1.00** | **0.00** |
| 9 | bernoulli | $0.45_{\pm0.17}$ | $0.70_{\pm0.08}$ | $0.90_{\pm0.05}$ | **1.00** | **1.00** | **0.00** |
| | dtd | $0.83_{\pm0.06}$ | $0.83_{\pm0.06}$ | $0.68_{\pm0.23}$ | **1.00** | **1.00** | **0.00** |
| | fashion mnist | $0.97_{\pm0.01}$ | $0.97$ | $0.16_{\pm0.04}$ | **1.00** | **1.00** | **0.01** |
| | gaussian | $1.00$ | $1.00$ | $0.01_{\pm0.01}$ | 1.00 | 1.00 | 0.00 |
| | omniglot | $0.99$ | $0.99$ | $0.06$ | 0.99 | 0.99 | 0.06 |
| | uniform | $0.72_{\pm0.15}$ | $0.82_{\pm0.09}$ | $0.50_{\pm0.21}$ | **1.00** | **1.00** | **0.00** |
| 15 | bernoulli | $0.18_{\pm0.06}$ | $0.55_{\pm0.02}$ | $0.99_{\pm0.01}$ | **1.00** | **1.00** | **0.00** |
| | dtd | $0.82_{\pm0.04}$ | $0.80_{\pm0.04}$ | $0.84_{\pm0.09}$ | **1.00** | **1.00** | **0.00** |
| | fashion mnist | $0.97$ | $0.97$ | $0.17_{\pm0.03}$ | **1.00** | **1.00** | **0.01** |
| | gaussian | $1.00$ | $1.00$ | $0.01_{\pm0.01}$ | 1.00 | 1.00 | $0.01_{\pm0.01}$ |
| | omniglot | $0.99$ | $0.99$ | $0.05$ | 0.99 | 0.99 | 0.05 |
| | uniform | $0.70_{\pm0.13}$ | $0.80_{\pm0.07}$ | $0.53_{\pm0.19}$ | **1.00** | **1.00** | **0.00** |

which the benefit of outlier exposure was the best were datasets of real images, and we think that in case unlabeled images with non-overlapping class distribution are available, using them can bring benefit.

**Effect of model capacity**   While the main experiments in the paper were conducted using large models, we also investigated whether ensembles of smaller models can benefit from our method. We followed the same $\lambda_M$ selection procedure as for the main experiments in the paper, and report results for PreResNet20 in Table C11 for an ensemble of size $M = 11$. We found that the optimal $\lambda_M$ in this case is smaller compared

Table C9: Walltime comparisons of Deep Ensembles and our method for $M = 11$ on CIFAR10 for 3 architectures. Both methods have been trained on 1 GPU for fair estimation of the computational overhead. The results reported over five runs with different random seeds.

| Architecture | Walltime (hours) | | Overhead |
|--------------|------------------|------|----------|
| | DE | Ours | |
| PreResNet164 | 18.81±0.18 | 32.81±0.25 | 74.39% |
| VGG16BN | 2.11±0.01 | 4.00±0.03 | 89.12% |
| WideResNet28x10 | 23.23±0.07 | 43.92±0.25 | 89.03% |

Table C10: Outlier exposure results for PreResNet164 trained on CIFAR100 with CIFAR10 as a weighting distribution for the diversity term. We found $\lambda_M = 1$ to be the best one for this experiment in the outlier exposure. The most optimal setting for our model was $\lambda_M = 3$ and $\lambda_M = 5$, respectively.

| OOD dataset | DE | | | Ours | | | Ours w. Outlier Exposure | | |
|---|---|---|---|---|---|---|---|---|---|
| | AUC (↑) | AP (↑) | FPR95 (↓) | AUC (↑) | AP (↑) | FPR95 (↓) | AUC (↑) | AP (↑) | FPR95 (↓) |
| bernoulli | $0.81_{\pm0.03}$ | $0.82_{\pm0.03}$ | $0.24_{\pm0.04}$ | **1.00** | **1.00** | **0.00** | **1.00** | **1.00** | 0.00 |
| blobs | $0.92_{\pm0.01}$ | 0.95 | $0.25_{\pm0.02}$ | $0.92_{\pm0.02}$ | $0.95_{\pm0.01}$ | $0.23_{\pm0.03}$ | $0.96_{\pm0.01}$ | 0.98 | $0.14_{\pm0.01}$ |
| dtd | $0.76_{\pm0.01}$ | $0.61_{\pm0.01}$ | $0.64_{\pm0.01}$ | $0.80_{\pm0.01}$ | $0.75_{\pm0.01}$ | $0.78_{\pm0.05}$ | $0.81_{\pm0.01}$ | $0.71_{\pm0.01}$ | 0.57 |
| gaussian | $0.80_{\pm0.01}$ | $0.83_{\pm0.01}$ | $0.37_{\pm0.02}$ | $0.95_{\pm0.02}$ | $0.97_{\pm0.01}$ | $0.16_{\pm0.05}$ | $0.94_{\pm0.01}$ | $0.96_{\pm0.01}$ | $0.19_{\pm0.03}$ |
| lsun | 0.86 | 0.81 | 0.45 | 0.87 | 0.85 | $0.47_{\pm0.01}$ | **0.89** | $0.88_{\pm0.01}$ | $0.40_{\pm0.01}$ |
| places | 0.82 | 0.77 | 0.52 | 0.83 | 0.81 | $0.60_{\pm0.03}$ | **0.86** | $0.84_{\pm0.01}$ | $0.49_{\pm0.01}$ |
| svhn | $0.80_{\pm0.01}$ | 0.96 | $0.55_{\pm0.01}$ | $0.83_{\pm0.01}$ | 0.96 | $0.50_{\pm0.01}$ | $0.78_{\pm0.01}$ | 0.95 | $0.56_{\pm0.02}$ |
| tiny imagenet | 0.82 | 0.79 | **0.53** | 0.82 | 0.79 | $0.58_{\pm0.01}$ | **0.83** | 0.79 | **0.52** |
| uniform | $0.51_{\pm0.09}$ | $0.66_{\pm0.05}$ | $0.55_{\pm0.09}$ | **1.00** | **1.00** | **0.00** | $0.98_{\pm0.01}$ | $0.98_{\pm0.02}$ | $0.05_{\pm0.02}$ |

to $\lambda_M$ for PreResNet164, and our method here does not yield any substantial boost over Deep Ensembles on these data. We found that all the models with small capacity trained on CIFAR diverged with high $\lambda_M$

Table C11: Out of distribution detection for a small-capacity model – PreResNet20 ($M = 11$). Results were averaged over 5 random seeds. Standard errors are not reported if they less than 0.01 across runs. DE indicates Deep Ensembles.

| Dataset | OOD dataset | DE | | | Ours | | |
|---|---|---|---|---|---|---|---|
| | | AUC (↑) | AP (↑) | FPR95 (↓) | AUC (↑) | AP (↑) | FPR95 (↓) |
| C10 | bernoulli | $0.99_{\pm0.01}$ | $0.99_{\pm0.01}$ | $0.05_{\pm0.02}$ | 1.00 | 1.00 | **0.00** |
| | blobs | 0.97 | 0.98 | 0.12 | 0.97 | 0.98 | $0.12_{\pm0.01}$ |
| | cifar100 | 0.88 | 0.86 | 0.37 | **0.89** | 0.86 | **0.36** |
| | dtd | 0.93 | $0.86_{\pm0.01}$ | $0.22_{\pm0.01}$ | $0.94_{\pm0.01}$ | $0.89_{\pm0.01}$ | $0.20_{\pm0.01}$ |
| | gaussian | $0.94_{\pm0.01}$ | $0.96_{\pm0.01}$ | $0.18_{\pm0.03}$ | $0.95_{\pm0.01}$ | $0.97_{\pm0.01}$ | $0.15_{\pm0.03}$ |
| | lsun | 0.93 | 0.92 | 0.25 | 0.94 | 0.93 | $0.23_{\pm0.01}$ |
| | places | 0.92 | 0.90 | 0.27 | **0.93** | **0.91** | 0.26 |
| | svhn | $0.91_{\pm0.01}$ | 0.98 | $0.23_{\pm0.01}$ | 0.91 | 0.98 | $0.23_{\pm0.01}$ |
| | tiny imagenet | 0.90 | 0.88 | 0.33 | 0.90 | 0.88 | **0.32** |
| | uniform | 0.99 | 0.99 | $0.03_{\pm0.01}$ | **1.00** | **1.00** | **0.00** |
| C100 | bernoulli | $0.91_{\pm0.05}$ | $0.93_{\pm0.04}$ | $0.19_{\pm0.08}$ | **1.00** | **1.00** | **0.00** |
| | blobs | 0.95 | 0.97 | $0.17_{\pm0.01}$ | **0.97** | **0.98** | 0.12 |
| | cifar10 | **0.78** | **0.73** | 0.63 | 0.77 | 0.72 | 0.63 |
| | dtd | $0.81_{\pm0.01}$ | $0.72_{\pm0.01}$ | $0.66_{\pm0.03}$ | $0.83_{\pm0.01}$ | $0.77_{\pm0.01}$ | $0.63_{\pm0.03}$ |
| | gaussian | $0.98_{\pm0.01}$ | 0.99 | $0.07_{\pm0.01}$ | 0.99 | 0.99 | $0.04_{\pm0.01}$ |
| | lsun | 0.88 | 0.87 | $0.43_{\pm0.01}$ | **0.89** | **0.88** | $0.41_{\pm0.01}$ |
| | places | 0.84 | 0.82 | $0.55_{\pm0.01}$ | **0.85** | **0.83** | $0.52_{\pm0.01}$ |
| | svhn | 0.85 | 0.97 | $0.48_{\pm0.01}$ | $0.84_{\pm0.01}$ | 0.97 | $0.49_{\pm0.02}$ |
| | tiny imagenet | 0.82 | 0.78 | 0.56 | 0.82 | **0.79** | **0.54** |
| | uniform | $0.90_{\pm0.03}$ | $0.92_{\pm0.02}$ | $0.23_{\pm0.07}$ | **1.00** | **1.00** | **0.00** |

**Robustness to distribution shift** As an additional evaluation, we investigated whether our method performs on par with DE under the distribution shift, to make sure that introduction of additional regularization did not affect the robustness properties. We thus use a corrupted version of CIFAR10 test set released by (Hendrycks & Dietterich, 2019), and it has been recently shown that DE outperform many other methods on this benchmark (Ovadia et al., 2019). Here, we report the results for VGG16BN and PreResNet164, as they yielded OOD performance gain in both SVHN and LSUN datasets. One can see from Figure C3 that our method performs on-par with DE, as no statistical significance in difference between methods can be concluded from this plot. This further supports our claims that the developed greedy ensemble training approach works the same or on par with DE.

**Effect of ensemble size on CIFAR** We ran our experiments using PreResNet164 with on CIFAR10/100, having $M \in \{3, 5, 7\}$ and $\lambda_M\{0.1, 0.5, 0.8, 1, 1.5, 2, 3\}$. Both in-domain accuracy, and the OOD detection on

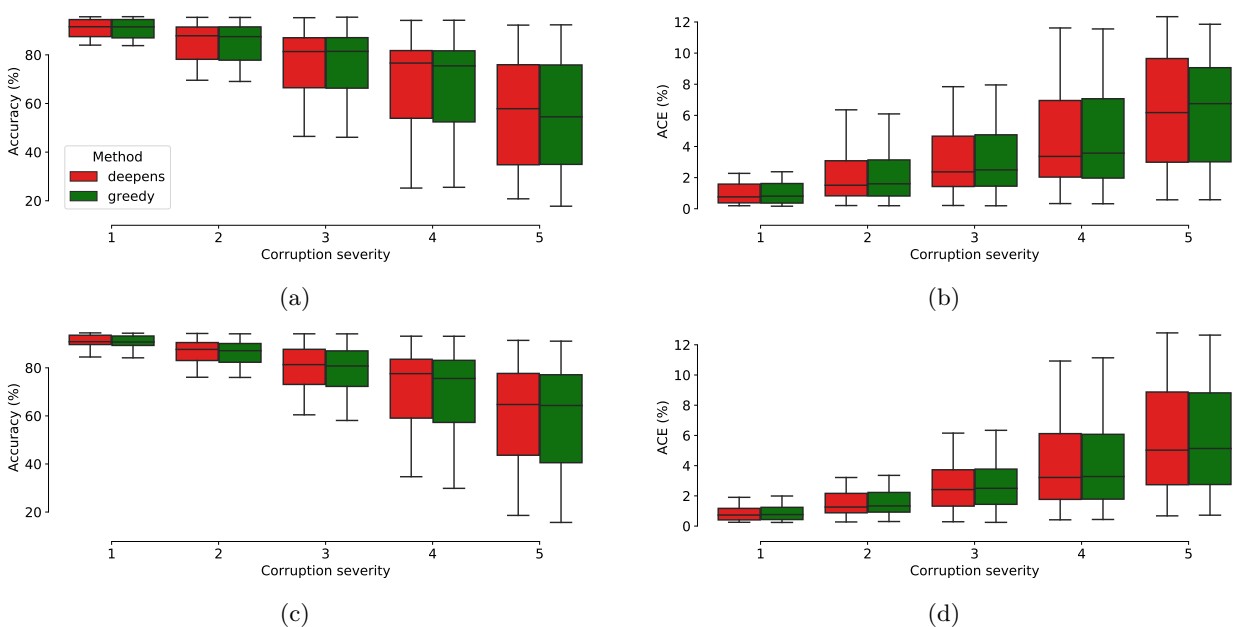

Figure C3: CIFAR10 Robustness benchmark results ($M = 11$) (Hendrycks & Dietterich, 2019). Subplots (a) and (b) show the results for PreResNet164 trained with $\lambda_M = 3$. Subplots (c) and (d) show the results for VGG16BN trained with $\lambda_M = 5$. The results have been averaged over 5 seeds.

LSUN and SVHN are shown in Table C12. The results in that table show that with small ensemble size, our method may yield better performance than DE on SVHN, abut even with an ensemble of size $M = 3$, it has a substantial boost over DE in detecting LSUN.

Table C12: Test set and OOD detection performances on PreResNet164 for ensemble sizes $M \in \{3, 5, 7\}$. We report the means over 5 different seeds. Standard errors are reported if they are non-zero across the runs.

| $M$ | Dataset | Method | Accuracy (%) | NLL $\times 100$ | ACE (%) | SVHN | | LSUN | |
|---|---|---|---|---|---|---|---|---|---|
| | | | | | | AUC ($\uparrow$) | AP ($\uparrow$) | AUC ($\uparrow$) | AP ($\uparrow$) |
| 3 | C10 | DE | $95.38_{\pm 0.04}$ | $15.59_{\pm 0.14}$ | $0.25_{\pm 0.01}$ | 0.93 | 0.95 | 0.92 | 0.87 |
| | | Ours | $95.32_{\pm 0.04}$ | $15.67_{\pm 0.11}$ | $0.24_{\pm 0.01}$ | $0.94_{\pm 0.01}$ | $0.96_{\pm 0.01}$ | **0.94** | $\mathbf{0.91_{\pm 0.01}}$ |
| | C100 | DE | $78.67_{\pm 0.04}$ | $84.35_{\pm 0.19}$ | 0.10 | $0.78_{\pm 0.02}$ | $0.87_{\pm 0.01}$ | 0.82 | $0.76_{\pm 0.01}$ |
| | | Ours | $78.51_{\pm 0.16}$ | $84.55_{\pm 0.76}$ | 0.10 | $0.76_{\pm 0.02}$ | $0.86_{\pm 0.01}$ | $0.82_{\pm 0.02}$ | $0.76_{\pm 0.02}$ |
| 5 | C10 | DE | $95.55_{\pm 0.03}$ | $14.14_{\pm 0.14}$ | $0.20_{\pm 0.01}$ | 0.93 | 0.96 | 0.92 | 0.88 |
| | | Ours | $95.58_{\pm 0.04}$ | $14.31_{\pm 0.07}$ | 0.19 | **0.94** | 0.96 | **0.95** | **0.93** |
| | C100 | DE | $79.50_{\pm 0.08}$ | $78.85_{\pm 0.28}$ | 0.09 | $0.78_{\pm 0.01}$ | $0.87_{\pm 0.01}$ | 0.84 | 0.79 |
| | | Ours | $79.33 \pm 0.13$ | $78.59_{\pm 0.33}$ | 0.09 | $0.80_{\pm 0.01}$ | 0.88 | **0.86** | $\mathbf{0.82_{\pm 0.01}}$ |
| 7 | C10 | DE | $95.64_{\pm 0.04}$ | $13.79_{\pm 0.06}$ | 0.19 | 0.94 | 0.96 | 0.92 | 0.88 |
| | | Ours | $95.62_{\pm 0.04}$ | $14.07_{\pm 0.19}$ | $0.22_{\pm 0.04}$ | 0.94 | 0.96 | **0.94** | $\mathbf{0.93_{\pm 0.01}}$ |
| | C100 | DE | $79.81_{\pm 0.06}$ | $76.12_{\pm 0.22}$ | 0.08 | $0.78_{\pm 0.01}$ | $0.87_{\pm 0.01}$ | 0.85 | 0.79 |
| | | Ours | $79.76_{\pm 0.08}$ | $78.09_{\pm 1.41}$ | $0.11_{\pm 0.02}$ | $0.78_{\pm 0.01}$ | $0.88_{\pm 0.01}$ | **0.86** | $\mathbf{0.83_{\pm 0.01}}$ |

**Qualitative results on CIFAR100.** In Figure C4, we further illustrate the capabilities of uncertainty estimation of our method for the PreResNet164 trained on CIFAR100 ($M = 11$). Our method has a better true positive rate (as can also be seen from the histograms), and slightly better precision when the threshold for the recall is high. We note that the histograms of epistemic uncertainties still overlap significantly, however,

with our method, the model does not have a high number of overconfident predictions coming from the OOD data anymore.

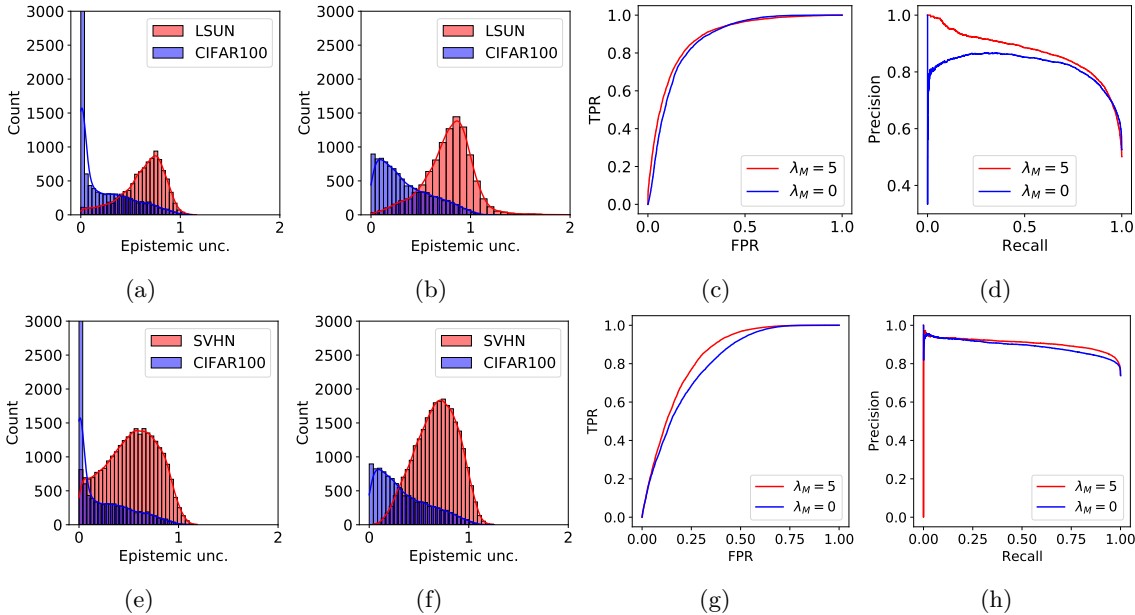

Figure C4: Uncertainty estimation quality on PreResNet164 (He et al., 2016) trained on CIFAR100 and evaluated on CIFAR100 test set vs LSUN and SVHN, respectively. Histograms (a) and (e) indicate Deep Ensembles (Lakshminarayanan et al., 2017). Histograms (b) and (f) show our method trained with $\lambda_M = 5$. Subplots (c) and (d) show the ROC and PR curves for the LSUN dataset, respectively. Subplots (g) and (h) show the ROC and PR curves for the SVHN dataset, respectively The curves were computed using average epistemic uncertainty per sample (5 seeds). Standard errors were < 0.01.

