# OpenReview forum: "Greedy Bayesian Posterior Approximation with Deep Ensembles"
_TMLR — Accepted by TMLR_

### Review · Reviewer_pi5e · 2022-04-12

**Summary Of Contributions:**

This paper studies the problem of constructing deep neural network ensembles. Unlike prior works that look at ensembles from the perspective of improving predictive accuracy, this papers focuses on recovering the true model posterior distribution. The authors measure the posterior recovery by f-divergence in the functional space. By a combinatorial analysis, the authors showed that the f-divergence is supermodular, which motivates a greedy method for selecting new models into the ensembles. The greedy method is implemented as a diversity-promoting regularization term, that encourages predictive difference on a set of input (the inputs could be sampled from a distribution but this distribution needs to be determined).

The author evaluates the proposed ensemble method by out-of-distribution detection evaluation and mainly compares to Deep Ensembles (2017). The proposed method demonstrates advantage over Deep Ensembles on out-of-distribution detection across all datasets that the authors have evaluated on. I found the empirical study to be comprehensive (if we include those in the appendix into cosideration). That said, I note that the proposed method does not provide accuracy improvement over Deep Ensembles.

Overall, this submission is stronger on the theory side, providing a new perspective to analyze neural network ensembles although the empirical gains seem less impressive.

**Broader Impact Concerns:**

I do not have any ethical concern with this paper. The proposed idea is general enough to be applied to many different applications.

**Requested Changes:**

- I would suggest moving alg.1 in the Appendix into the main paper to highlight the actual algorithm used for the reported experiments. The current separation makes it quite confusing.

**Strengths And Weaknesses:**

Strength:
- The proposed idea of taking a functional view of the ensembles is novel and technically interesting.
- The authors motivates the greedy selection algorithm by an analysis on submodular optimization. I think this makes meaningful efforts toward understanding the guarantees of practical algorithms, which I appreciate.
- The paper is well-written and organized in a meaningful way.

Weakness:
- the weighting function, which essentially defines the diversity term, is heuristically defined and could have important impact on the practical performance. In the current experimental setup, the authors fit “a Gaussian to every data dimension with the variance x5 larger than the original variance”. I question the generalizability of such construction. I would imagine this to work better for regular data such as images, but can it work for audio, or text data? Because of this, I think the applicability of the current practical algorithm is limited.
- I appreciate that the authors candidly report the issue with additional computational overhead. Can the authors report concrete wall clock time comparison with other comparable methods? say deep ensemble? The reported experiments are done with a 11 model ensemble. Does this mean a 11x slowdown to the overall run time? If so, this could be quite prohibitive.

Nit:
- page 4, “we, however, that in practice” → “we, however, believe that”?

---

> ### Author Response · Authors · 2022-04-29
> **Authors' response to pi5e**
>
> **====Strengths====**
>
> **Comment:**
> 1. The proposed idea of taking a functional view of the ensembles is novel and technically interesting.
> 2. The authors motivates the greedy selection algorithm by an analysis on submodular optimization. I think this makes meaningful efforts toward understanding the guarantees of practical algorithms, which I appreciate.
> 3. The paper is well-written and organized in a meaningful way.
>
> **Reply:** Thanks for these comments!
>
> **====Weaknesses====**
>
> **Comment:** the weighting function, which essentially defines the diversity term, is heuristically defined and could have important impact on the practical performance. In the current experimental setup, the authors fit “a Gaussian to every data dimension with the variance x5 larger than the original variance”. I question the generalizability of such construction. I would imagine this to work better for regular data such as images, but can it work for audio, or text data? Because of this, I think the applicability of the current practical algorithm is limited.
>
> **Reply:** We agree with the reviewer, and we have made an ablation study (Table C9; Appendix C4). The main insight from that table, is that one can also use images, however, this did not seem to provide any substantial benefits over the proposed heuristic. We do not wish to speculate on all possible types of data, but our experience with images indicates that an operation to generate samples beyond the training dataset in the diversity term is useful.  We expect that such operations can be generated for other domains by e.g. increasing the variance in the data generation procedure of a (deep) generative model.  Applications to audio, text, etc. are interesting areas for future research.
>
> **Comment:** I appreciate that the authors candidly report the issue with additional computational overhead. Can the authors report concrete wall clock time comparison with other comparable methods? say deep ensemble? The reported experiments are done with a 11 model ensemble. Does this mean a 11x slowdown to the overall run time? If so, this could be quite prohibitive.
>
> **Reply:** We have noted that some degree of parallelization is possible through parallel submodular maximization algorithms.  Thus, training time is sub-linear in the number of models, but not embarrassingly parallel as is the case for deep ensembles. This is the price to pay for better uncertainty estimation.  Test time does not depend on the training procedure, and is not more expensive than other deep ensemble methods. We have added relevant discussion and reported the exact walltimes (Table C9) for the main experiments on CIFAR10.
>
> **====Typos and other changes====**
>
> **Comment:** page 4, “we, however, that in practice” → “we, however, believe that”?
>
> **Reply:** Fixed.
>
> **Comment:** I would suggest moving alg.1 in the Appendix into the main paper to highlight the actual algorithm used for the reported experiments. The current separation makes it quite confusing.
>
> **Reply:** Fixed.

---

### Review · Reviewer_Ug4Q · 2022-04-18

**Summary Of Contributions:**

This paper presents a new angle on the question of Bayesian interpretation of deep ensembles by deriving and proposing a new approach that considers minimising an f-divergence between the true posterior and a function space KDE of the ensemble of fixed cardinality (ie number of members of the ensemble). Several approximations are derived to maintain computational tractability, including a greedy construction as well as using a diversity term that is approximated via MC integration on a weighting distribution p* on inputs (which is just Gaussian in input space). Experiments are provided comparing the new method to standard deep ensembles (Lakshminarayanan et al 2017) demonstrating improved performance in terms of uncertainty quantification and ood detection, both on two moons and on image datasets.

**Requested Changes:**

Besides the requested changes above, I have the following more minor suggestions:

1. yeilds -> yields
2. change V -> Z in def 1 for consistency.
3. "We, however, that in practice" needs rewording.
4. Bring the main algorithm used in practice (Alg B.2 i believe) into the main paper, rather than the algorithm that is currently in the main paper which isn't used (Alg. 1).
5. Please provide more justification for the mean-field approximation in Eq 13. I realise it may be convenient computationally, but do the authors have any understanding of how loose it is/settings where it will be tighter?
6. Please cite missed related works on Bayesian interpretations to Deep Ensembles https://openreview.net/forum?id=BJlahxHYDS and https://arxiv.org/abs/2007.05864

**Strengths And Weaknesses:**

Strengths:

1. The use of general f-divergence as an optimisation algorithm for an ensemble of Ns is novel as far as I am aware and should be of interest to some in the TMLR community.
2. The authors provide some theoretical analysis to support their claims (e.g. Theorem 1 and Proposition 1), though see weaknesses below for some comments on the relevance of the theory.
3. The experiments seem to demonstrate benefits of the proposed method in terms of uncertainty quantification relative to deep ensembles.
4. The related work is largely thorough (though see requested changes for some missed works).
5. The authors provide a limitations section.

Weaknesses:

1. Clarity: the paper could be improved in terms of writing. Besides typos (in requested changes), there are several claims that are largely unsubstantiated imo that ought to be developed/clarified or accompanied by citations, such as:

a. "DE... has arbitrar(il)y bad approxmation guarantees"\
b. "the performance of those methods is not state-of-the-art due to the use of BNN priors"\
c. "it has been shown experimentally that every ensemble member may discover different nodes of the posterior distribution in the function space".\
d. "This algorithm has approximation guarantee of 1/e in general."

There are other situations where the writing is detrimental to clarity beyond simple typos. E.g:

e. "marginal gain of the total objective" (neither of these two terms are defined yet) in the abstract\
f. p(z|D) is introduced before sec 2.2. before any notation concerning Bayes' rule in sec 4. \
g. likewise, "mean-seeking behaviour" is introduced without real definition just before sec 2.2, what does this mean? I see in sec 3 there is a semi-definition: 'cover the posterior distribution as much as possible', but this is not particularly precise to me. \
h. In line 6 of Alg 1., |T|=M-m right not M?\
i. Equation 12 should it be 1/(k-1) not 1/M?\
j. Equation 14 should it be *negative* marginal gain, so that minimising the objective J leads to maximising marginal gain?\
k. I'm not really sure in the sentence between eq 4 and eq 5 why there is a d in the definition of q_M(z), and also the index is from m not j\
l. "Minimization of (5) is equivalent..." it is not clear what we are minimising over, I take it that it is z_m but you should state that p(z) is fixed (and indeed is the posterior in your case).

2. Experiments on accuracy: it is shown that the proposed methods improve vs Deep Ensembles in terms of uncertainty quantification. However, ensembles also improve accuracy in NNs https://arxiv.org/abs/2012.09816, does the proposed methods also improve accuracy? Moreover are the authors able to provide experiments detailing how the size of the ensemble affects the improvement in uncertainty/accuracy, it seems the experiments provided all set ensemble size M=11.

3. Motivation: The motivation for why one would want to take a Bayesian interpretation for deep ensembles is somewhat lacking imo. The authors write: "none of these (other) works takes the perspective of approximation the Bayesian posterior", but the authors do not substantiate this statement for why one would want to do so. Moreover, I'm not sure if I properly understood section 3: is the point of introducing submodular analysis and theorem 1 because minimising f-diverfences benefit from approximation guarantees as a result? If so, this is quite confusion giving the bottom half of page 4 suggests it is difficult to approximate such optimisation problems. If not, then I am unclear what the motivation for section 3 is.

4. The authors write '"the impact of the choice of function space... is therefore unavoidable when designing algorithms for approximation the Bayesian posterior". This is likely true, but the justification for this is only that the *upper bound* in eq (8) could be bad, whereas there is no such statement for the true quanitity we care about.

5. Lack of parallelisability: I commend the authors for pointing this out. However, even the fix that in Ene and Nguyen 2020 requires communication between members of the ensemble trained in parallell for the diversity promoting term, which Deep ensembles do not. However, assuming communcation is possible, this would enable parallel training OTOH, the proposed method here requires the members of ther ensemble to be trained sequentially (i believe, looking at Alg B2), which is worse.

---

> ### Author Response · Authors · 2022-04-29
> **Authors' response to Ug4Q (part 1)**
>
> **==== Strengths ====**
>
> **Comment:**
> 1. The use of general f-divergence as an optimisation algorithm for an ensemble of Ns is novel as far as I am aware and should be of interest to some in the TMLR community.
> 2. The authors provide some theoretical analysis to support their claims (e.g. Theorem 1 and Proposition 1), though see weaknesses below for some comments on the relevance of the theory.
> 3. The experiments seem to demonstrate benefits of the proposed method in terms of uncertainty quantification relative to deep ensembles.
> 4. The related work is largely thorough (though see requested changes for some missed works).
> 5. The authors provide a limitations section.
>
> **Reply:** Thank you for these comments.
>
> **Comment:** Clarity: the paper could be improved in terms of writing. Besides typos (in requested changes), there are several claims that are largely unsubstantiated imo that ought to be developed/clarified or accompanied by citations, such as:
> 1. "DE... has arbitrar(il)y bad approxmation guarantees"
> 2. "the performance of those methods is not state-of-the-art due to the use of BNN priors"
> 3. "it has been shown experimentally that every ensemble member may discover different nodes of the posterior distribution in the function space".
> 4. "This algorithm has approximation guarantee of 1/e in general."
>
> **Reply:**
> 1. Please see the example in the next sentence, which we have now made even clearer: “For example, the resulting approximation can be poor in the case when the true posterior distribution is unimodal, skewed and long-tailed.”
> 2. The BNN priors are conventionally defined only over weights, and techniques such as batch normalization are never used, since they have no Bayesian interpretation. Naturally, this can be seen as enforcing a prior on a neural network architecture. We have clarified this claim in the text.
> 3. Missing reference has been provided
> 4. Missing reference added
>
> **Comment:** There are other situations where the writing is detrimental to clarity beyond simple typos. E.g:
> 1. "marginal gain of the total objective" (neither of these two terms are defined yet) in the abstract
> 2. p(z|D) is introduced before sec 2.2. before any notation concerning Bayes' rule in sec 4.
> 3. likewise, "mean-seeking behaviour" is introduced without real definition just before sec 2.2, what does this mean? I see in sec 3 there is a semi-definition: 'cover the posterior distribution as much as possible', but this is not particularly precise to me.
> 4. In line 6 of Alg 1., |T|=M-m right not M?
> 5. Equation 12 should it be 1/(k-1) not 1/M?
> 6. Equation 14 should it be negative marginal gain, so that minimising the objective J leads to maximising marginal gain?
> 7. I'm not really sure in the sentence between eq 4 and eq 5 why there is a d in the definition of q_M(z), and also the index is from m not j
> 8. "Minimization of (5) is equivalent..." it is not clear what we are minimising over, I take it that it is z_m but you should state that p(z) is fixed (and indeed is the posterior in your case).
>
> **Reply:**
> 1. The wording is now more precise and elaborate in the abstract: “Subsequently, we consider the problem of greedy ensemble construction. From the marginal gain on the negative $f$-divergence, which quantifies an improvement in posterior approximation yielded by adding a new component into the KDE”
> 2. We have introduced a paragraph in the introduction, which concerns Bayes’ rule.
> 3. We have elaborated better what mean- and mode-seeking behavior are.
> 4. No, please see [1], which introduced the random greedy algorithm
> 5. No, it is not the case. 1/M is a constant known in advance, and stays fixed. This is one of our insights, which allows the application of the random greedy algorithm.
> 6. We have fixed the typos everywhere in the text. Thanks a lot for pointing this out. Indeed, (14) is the negative marginal gain
> 7. The indexing has been fixed (also in response to another reviewer).  Thank you for catching this. We have removed the dependence on d here as symmetry and translation invariance are not required at this point, and have introduced the distance between functions in Section 4.1, consistent with the restructuring of the text in response to another reviewer.
> 8. We have added the clarification in the text specifying that the minimization is w.r.t. $q_M(z)$ and that $p(z)$ is a fixed distribution.

---

> > ### Author Response · Authors · 2022-04-29
> > **Authors' response to Ug4Q (part 2)**
> >
> >
> > **Comment:** Experiments on accuracy: it is shown that the proposed methods improve vs Deep Ensembles in terms of uncertainty quantification. However, ensembles also improve accuracy in NNs https://arxiv.org/abs/2012.09816, does the proposed methods also improve accuracy? Moreover are the authors able to provide experiments detailing how the size of the ensemble affects the improvement in uncertainty/accuracy, it seems the experiments provided all set ensemble size M=11.
> >
> > **Reply:** We answer to this comment in parts:
> > 1. Our method does not aim to provide better accuracy, and its only aim is to get better coverage of the true posterior in the resulting approximation. Furthermore, as highlighted in Appendix C1, we used in-domain test-set accuracy to find the ensemble that has at least the same accuracy before evaluating it on the OOD benchmark. As stated in our manuscript, we did not observe (as it was not in the objective function) any accuracy gain.
> > 2. The results on different ensemble sizes have been shown in Table C8, Appendix C4 (also in the submitted version) for CIFAR and for MNIST in Tables C7 and C8
> >
> > **Comment:** Motivation: The motivation for why one would want to take a Bayesian interpretation for deep ensembles is somewhat lacking imo. The authors write: "none of these (other) works takes the perspective of approximation the Bayesian posterior", but the authors do not substantiate this statement for why one would want to do so.
> >
> > **Reply:** A new 2nd paragraph was added to the introduction, which gives a better perspective why one does need a Bayesian approach.
> >
> > **Comment:** Moreover, I'm not sure if I properly understood section 3: is the point of introducing submodular analysis and theorem 1 because minimising f-diverfences benefit from approximation guarantees as a result? If so, this is quite confusion giving the bottom half of page 4 suggests it is difficult to approximate such optimisation problems. If not, then I am unclear what the motivation for section 3 is.
> >
> > **Reply:** The point of introducing section 3 was to approach the problem of approximating a posterior distribution from a perspective of minimizing an f-divergence between the posterior, and the KDE. We highlighted that the f-divergence is supermodular with respect to a KDE, and thus its minimization is a non-monotone submodular maximization problem. Combinatorial problems of this type can be solved using generic algorithms, which have approximation guarantees.
> >
> > **Comment:** The authors write '"the impact of the choice of function space... is therefore unavoidable when designing algorithms for approximation the Bayesian posterior". This is likely true, but the justification for this is only that the upper bound in eq (8) could be bad, whereas there is no such statement for the true quanitity we care about.
> >
> > **Reply:** We have made the following changes to address this comment and comments from another reviewer.
> > 1. The general set-theoretic findings were disentangled from the neural network-related statements in order to improve the clarity of the paper.
> > 2. The claim is now stated w.r.t. the ground set V.
> >
> > Statements about the importance of the complexity of a function space based on upper bounds are well established, e.g. in statistical learning theory, and frequently used to justify regularization or choice of a function class as minimizing an upper bound.  The bound in Eq (8) (new numbering) has similar implications.
> >
> > **Comment:** Lack of parallelisability: I commend the authors for pointing this out. However, even the fix that in Ene and Nguyen 2020 requires communication between members of the ensemble trained in parallell for the diversity promoting term, which Deep ensembles do not. However, assuming communcation is possible, this would enable parallel training OTOH, the proposed method here requires the members of ther ensemble to be trained sequentially (i believe, looking at Alg B2), which is worse.
> >
> > **Reply:** We understand the concern of the reviewer, however, our work does not claim that we propose an alternative to Deep Ensembles, which is as efficient computationally. Instead, our main claims lie in the domain of understanding them from a combinatorial perspective, and outperforming them in OOD detection. Since we have mentioned the limitation of our method already, we do not consider this to be an issue to be corrected in the manuscript.

---

> > > ### Author Response · Authors · 2022-04-29
> > > **Authors' response to Ug4Q (part 3)**
> > >
> > > **====Typos and missing citations====**
> > >
> > > **Comment:** Besides the requested changes above, I have the following more minor suggestions:
> > > 1. yeilds -> yields
> > > 2. change V -> Z in def 1 for consistency.
> > > 3. "We, however, that in practice" needs rewording.
> > > 4. Bring the main algorithm used in practice (Alg B.2 i believe) into the main paper, rather than the algorithm that is currently in the main paper which isn't used (Alg. 1).
> > > 5. Please provide more justification for the mean-field approximation in Eq 13. I realise it may be convenient computationally, but do the authors have any understanding of how loose it is/settings where it will be tighter?
> > > 6. Please cite missed related works on Bayesian interpretations to Deep Ensembles https://openreview.net/forum?id=BJlahxHYDS and https://arxiv.org/abs/2007.05864
> > >
> > > **Reply:**
> > > 1. Fixed
> > > 2. We have fixed the notation. Z is a solution, V is the ground set.
> > > 3. Fixed.
> > > 4. Fixed
> > > 5. We have improved this section with regard to the mean field approximation. Specifically, we introduced it now already in the general f-divergence section. Furthermore, we added additional text about the reasons why we chose this approach. We have not made any statement with regard to its tightness, and we have not seen any work defining it.
> > > 6. We have mentioned those papers in the introduction.
> > >
> > > **====References====**
> > >
> > > [1] Buchbinder, N., Feldman, M., Naor, J., & Schwartz, R. (2014, January). Submodular maximization with cardinality constraints. In Proceedings of the twenty-fifth annual ACM-SIAM symposium on Discrete algorithms (pp. 1433-1452). Society for Industrial and Applied Mathematics.

---

### Review · Reviewer_qSkN · 2022-04-20

**Summary Of Contributions:**

Consider the deep ensembles approach to predictive uncertainty estimation. This paper proposes a heuristic greedy algorithm that trains several neural networks via minimizing the averaged empirical risk with a diversity-promoting regularizer. The proposed algorithm is inspired by the observation that any f-divergence between a probability distribution and normalized mixture of kernels is supermodular (Theorem 1) and the forward greedy selection algorithm in submodular optimization.

**Broader Impact Concerns:**

None.

**Requested Changes:**

Please address the weaknesses above.

Typos:
- First line of A.1: proposition -> theorem.
- Line following (4): *where* $K_j ( z )$ ...
- p. 4: A word is missing between "however, " and "that in practice."
- Proof of Proposition 1 on p. 5: *in* Appendix A.2.
- Line above (11): simplifies -> becomes. (There is nothing simplified.)
- Line above (14): at *the* $k^{\text{th}}$
- Line above (25): *the* set function

**Strengths And Weaknesses:**

==== Strengths ====

1. The combinatorial perspective is fresh.

2. Theorem 1 and its proof are interesting. Theorem 1 regarding the supermodularity of f-divergences is fundamental yet seems to be novel. The proof exploits a characterization of convexity I did not know before.

The paper highlights that the algorithm regards the function space instead of the parameter space. But the proposed solution ((15) with the explanations above it) looks naive.

==== Weakness  ====

Regarding the idea, I have the following confusions.

1. There is a huge gap between the combinatorial insight and the actual implementation. Algorithm 1, the forward greedy selection algorithm, requires a finite ground set as the input. What is the analogue of the finite ground set in the deep ensembles case?

2. Why the sampling procedure in Line 7 & 8 of the forward greedy selection algorithm corresponds to training a neural network with random initialization needs more explanations.

3. A derivation of (14) should be provided.

The presentation needs to be improved.

1. To me, a theoretician unfamiliar with Bayesian deep learning and submodular optimization, the presentation is difficult to digest. Below are some examples. In particular, terms like "predictive uncertainty", "deep ensembles", and "mode seeking" appear without definitions, forbidding non-experts to quickly digest this paper.

2. The mathematical writing is imprecise and careless. There are a lot of instances, such as:
    - In (1), the constraint $Z \subset \mathcal{F}$ is missing.
    - In (4), the set on which the integration is done is missing.
    - In the definition of $q_M (z)$, the subscript should be $m$ instead of $j$.
    - In the definition of $C$, the subscript $f$ of $D$ is missing.
    - Proof of Theorem 1: The condition that $a, b > 0$ is missing.
    - (22): It should be that the function is *convex on $[a, b]$* instead of that for any input in $[a, b]$ the function is convex.
    - Line following (22): $\frac{1}{\alpha} y$ is a quantity instead of a mapping.
    - (24): The word "convex" is missing on the left-hand side.

3. The proposed algorithm also looks imprecise and careless.
    - The definitions of "set_seed" and "random_init" are not given.
    - In Line 9, $\theta_m$ is chosen randomly (my guess), but then in Line 10, it is replaced by the solution of an optimization problem. What is the point of randomly choosing $\theta_m$ then?
    - I guess what Line 9 & 10 want to convey is to train the neural network with random initialization, but I am not sure.
    - It is an abuse of notation to write $\theta_m$ for both the solution of the optimization problem and the variable to be optimized.

3. Discussion on using the randomized initialization heuristic should not appear in Section 3.2, as that section only considers the problem of minimizing an f-divergence. The discussion should be moved to Section 4.

4. The proposed algorithm (Algorithm B2) should appear in the main text instead of the appendix. Indeed, Section 4 gives the proposed algorithm, but it looks too terse.

---

> ### Author Response · Authors · 2022-04-29
> **Authors' response to qSkN (part 1)**
>
> **==== Strengths ====**
>
> **Comment:**
> 1. The combinatorial perspective is fresh.
> 2. Theorem 1 and its proof are interesting. Theorem 1 regarding the supermodularity of f-divergences is fundamental yet seems to be novel. The proof exploits a characterization of convexity I did not know before.
>
> **Reply:** Thanks, we appreciate that the reviewer values the theoretical insights presented in the paper!
>
>
> **Comment:** The paper highlights that the algorithm regards the function space instead of the parameter space. But the proposed solution ((15) with the explanations above it) looks naive.
>
> **Reply:** We believe the solution in (15) is very natural, based on Monte Carlo integration of the canonical function norm under the measure defined by p(x).  The NP-hardness result in Rannen-Triki et al. indicates that there is likely not a “clever” solution that is significantly better than this one.  It is not clear to us what solution here would not be “naive” or indeed what that term means to the reviewer in this context.
>
>
> **==== Weakness ====**
>
>
> **Comment:** There is a huge gap between the combinatorial insight and the actual implementation. Algorithm 1, the forward greedy selection algorithm, requires a finite ground set as the input. What is the analogue of the finite ground set in the deep ensembles case?
>
> **Reply:** Thanks for pointing this out. We have made it clear that the ground set is a class of continuous functions realizable on a computer. As the set of functions realizable by fixed precision weights is indeed finite, this does not pose any problem in practice - the theory holds when the set is arbitrarily large.
>
>
> **Comment:** Why the sampling procedure in Line 7 & 8 of the forward greedy selection algorithm corresponds to training a neural network with random initialization needs more explanations.
>
> **Reply:** Thanks for this comment.  We have improved the explanation of this step in the paper, which in the revised version can be found in Section 4.2 paragraph “The resulting algorithm.”  A short version of the justification of the approach is that the stochasticity of the optimization procedure is necessarily at least as large as any randomization that would be introduced by Algorithm 1 line 7.
>
>
> **Comment:** A derivation of (14) should be provided.
>
> **Reply:** We have added additional details in the paragraph before Equation (15) (new numbering) indicating that the objective is the result of a drop-in replacement of the approximation in Equation (13) (new numbering), which builds the bridge between the posterior definition and the objective J.
>
>
> **Comment:** The presentation needs to be improved.
>
> **Reply:**  We thank the reviewer for the provided comments. We have improved the presentation by addressing each of them.
>
>
> **Comment:** To me, a theoretician unfamiliar with Bayesian deep learning and submodular optimization, the presentation is difficult to digest. Below are some examples. In particular, terms like "predictive uncertainty", "deep ensembles", and "mode seeking" appear without definitions, forbidding non-experts to quickly digest this paper.
>
> **Reply:** We have added in text better elaboration about mean- and mode-seeking.
>
> **Comment:** The mathematical writing is imprecise and careless.
>
> **Reply:** We appreciate the detailed comments. Please see our responses below.
>
>
> **Comment:** In (1), the constraint $Z \subset \mathcal{F}$  is missing.
>
> **Reply:** We fixed this, by introducing the ground set $V$, and specifying it in (1). We have refactored the mathematical presentation to talk simply about a ground set with probability measure through the section on f-divergences, and then developing its application to neural networks.
>
>
>
>
>
> **Comment:** In (4), the set on which the integration is done is missing.
>
> **Reply:** We have adjusted our definition of an $f$-divergence, mentioning a measurable space, and the base measure, with respect to which the integration is done. Thie integration set is now specified.
>
>
> **Comment:** In the definition of $q_M (z)$, the subscript should be $m$ instead of $j$.
>
> **Reply:** Fixed.
>
>
> **Comment:** In the definition of $C$, the subscript  $f$ of $D$ is missing.
>
> **Reply:** Thanks, fixed.
>
>
> **Comment:** Proof of Theorem 1: The condition that a, b > 0 is missing.
>
> **Reply:** This is not necessary. Convexity holds regardless where it is on a real line. We have specified that x1>0 in the line before (19) (old equation numbering) thus making clear that there is no divide-by-zero or change in sign.
>
>
> **Comment:** (22): It should be that the function is convex on [a, b]  instead of that for any input in [a, b] the function is convex.
>
> **Reply:** Fixed.
>
>
> **Comment:** Line following (22): $\frac{1}{\alpha} y$ is a quantity instead of a mapping.
>
> **Reply:** We have changed the presentation to emphasize that this is a reparameterization.

---

> > ### Author Response · Authors · 2022-04-29
> > **Authors' response to qSkN (part 2)**
> >
> >
> > **Comment:** (24): The word "convex" is missing on the left-hand side.
> >
> > **Reply:** Fixed
> >
> >
> > **Comment:** The proposed algorithm also looks imprecise and careless.
> >
> > **Reply:** See the comment below.
> >
> >
> > **Comment:** The definitions of "set_seed" and "random_init" are not given.
> >
> > **Reply:** Fixed in the algorithm caption.
> >
> >
> > **Comment:** In Line 9, $\theta_m$ is chosen randomly (my guess), but then in Line 10, it is replaced by the solution of an optimization problem. What is the point of randomly choosing  then?
> >
> > **Reply:** We agree with the reviewer and have updated the algorithm and the description accordingly.
> >
> >
> > **Comment:** It is an abuse of notation to write $\theta_m$ for both the solution of the optimization problem and the variable to be optimized.
> >
> > **Reply:** Fixed.
> >
> >
> > **Comment:** Discussion on using the randomized initialization heuristic should not appear in Section 3.2, as that section only considers the problem of minimizing an f-divergence. The discussion should be moved to Section 4.
> >
> > **Reply:** Fixed
> >
> >
> > **Comment:** The proposed algorithm (Algorithm B2) should appear in the main text instead of the appendix. Indeed, Section 4 gives the proposed algorithm, but it looks too terse.
> >
> > **Reply:** Fixed.
> >
> >
> > **====Typos====**
> >
> >
> > **Comment:** First line of A.1: proposition -> theorem.
> >
> > **Reply:** Fixed
> >
> >
> > **Comment:** Line following (4): where $K_j ( z )$ …
> >
> > **Reply:** Fixed
> >
> >
> > **Comment:** p. 4: A word is missing between "however, " and "that in practice."
> >
> > **Reply:** Fixed
> >
> >
> > **Comment:** Proof of Proposition 1 on p. 5: in Appendix A.2.
> >
> > **Reply:** Fixed
> >
> >
> >
> > **Comment:** Line above (11): simplifies -> becomes. (There is nothing simplified.)
> >
> > **Reply:** Fixed
> >
> >
> > **Comment:** Line above (14): at the $k^th$
> >
> > **Reply:** Fixed
> >
> >
> > **Comment:** Line above (25): the set function
> >
> > **Reply:** Fixed

---

> > > ### Comment · Reviewer_qSkN · 2022-05-21
> > > **My comments after reading the author response**
> > >
> > > This work is interesting to me regarding the correspondence between Algorithm 1 (random greedy algorithm) and Algorithm 2 (implementation). Indeed, Algorithm 2 can be naturally interpreted as a very direct greedy algorithm; interestingly, by a seemingly naive argument leveraging the finite precision nature of computers, Algorithm 2 is connected to Algorithm 1 that has a rigorous approximation ratio guarantee. There is an obvious gap between the two algorithms though; Algorithm 2 is not equivalent to Algorithm 1.
> > >
> > > Algorithm 2 only applies to one specific diversity promoting regularization function. This is a significant limitation.
> > >
> > > The 1/e approximation ratio guarantee of Algorithm 1 only applies to the case where the cardinality is upper bounded by M, instead of that it is exactly M, if I understand the paper by Buchbinder et al. correctly.

---

> > > > ### Author Response · Authors · 2022-05-25
> > > > **Reply to reviewer qSkN**
> > > >
> > > > We thank the reviewer for the provided comments. Below are our responses, and the newest revision has been uploaded.
> > > >
> > > > **Comment:**
> > > > This work is interesting to me regarding the correspondence between Algorithm 1 (random greedy algorithm) and Algorithm 2 (implementation). Indeed, Algorithm 2 can be naturally interpreted as a very direct greedy algorithm; interestingly, by a seemingly naive argument leveraging the finite precision nature of computers, Algorithm 2 is connected to Algorithm 1 that has a rigorous approximation ratio guarantee. There is an obvious gap between the two algorithms though; Algorithm 2 is not equivalent to Algorithm 1.
> > > >
> > > > **Reply:** We agree with the reviewer and have added a discussion paragraph (highlighted in blue). We have also highlighted in the description of Algorithm 2 that it is a “Random Greedy-based” algorithm (highlighted in blue)
> > > >
> > > > **Comment:** Algorithm 2 only applies to one specific diversity promoting regularization function. This is a significant limitation.
> > > >
> > > > **Reply:** We have added this limitation as a part of the new discussion paragraph.
> > > >
> > > > **Comment:** The 1/e approximation ratio guarantee of Algorithm 1 only applies to the case where the cardinality is upper bounded by M, instead of that it is exactly M, if I understand the paper by Buchbinder et al. correctly.
> > > >
> > > > **Reply:** Thanks for catching this! We have fixed the type to highlight that it is approximately 1/e.

---

### Comment · Action_Editors · 2022-04-26
**Thank you everyone**

Thank you everyone for your detailed reviews!

We will now start the discussion period - please ask any clarifying questions, and the authors are encouraged to respond to the reviews eg to clarify points of confusion etc.

The goal of the discussion period is for the reviewers to gather all the information needed to be comfortable submitting a decision recommendation for this submission within 2 weeks.

---

> ### Author Response · Authors · 2022-04-29
> **Responses to the reviews**
>
> We thank the reviewers for the provided detailed comments. We respond to every one of them in the corresponding threads.

---

> > ### Author Response · Authors · 2022-05-02
> > **Changes are also now reflected in the submitted PDF**
> >
> > Dear all,
> >
> > We have noticed that OpenReview's tool for pdf comparison is unable to compare the revision with the first submitted version. We have thus marked the changes in red ourselves and re-submitted the manuscript. We hope that this will ease the review process.
> >
> > All the best,
> > TMLR Paper 8 Authors

---

### Comment · Action_Editors · 2022-05-12
**Official recommendations**

Dear reviewers,

Could I please ask you to submit your official recommendation for the submission, within the next week? (by May 18).

Note that acceptance criteria for TMLR are technical correctness and clarity of presentation, rather than significance or impact. This is mostly assessed by determining whether the claims made in the submission are supported by accurate, convincing and clear evidence. An implication of the above is that, in a situation where some of the claims made by a submission aren't backed by sufficient evidence, instead of asking the authors to provide new evidence, you may instead ask that the authors reduce or adjust the claims made in the submission.

AE

---

### Decision · Action_Editors · 2022-05-27

**Recommendation:** Accept as is

**Comment:**

Reviewers are happy with the submission and recommenced accept.

---

> ### Author Response · Authors · 2022-06-01
> **Camera ready version has been uploaded**
>
> We have de-anonymized our submission and uploaded the final version of the paper.
>
> Best wishes,
> - The authors.